# Zero-Shot Detection of LLM-Generated Text via Implicit Reward Model

**Runheng Liu, Heyan Huang, Xingchen Xiao, Zhijing Wu**[*]
School of Computer Science and Technology, Beijing Institute of Technology
{rhliu,hhy63,xcxiao,zhijingwu}@bit.edu.cn

## Abstract

Large language models (LLMs) have demonstrated remarkable capabilities across various tasks. However, their ability to generate human-like text has raised concerns about potential misuse. This underscores the need for reliable and effective methods to detect LLM-generated text. In this paper, we propose IRM, a novel zero-shot approach that leverages Implicit Reward Models for LLM-generated text detection. Such implicit reward models can be derived from publicly available instruction-tuned and base models. Previous reward-based method relies on preference construction and task-specific fine-tuning. In comparison, IRM requires neither preference collection nor additional training. We evaluate IRM on the DetectRL benchmark and demonstrate that IRM can achieve superior detection performance, outperforms existing zero-shot and supervised methods in LLM-generated text detection.

## 1 Introduction

Large Language Models (LLMs) have demonstrated remarkable capabilities across various tasks [1, 2]. In particular, their ability to follow instructions and generate human-like content has made them essential tools in various real-world applications. However, this trend raises concerns about the misuse of LLMs, such as generating fake news [3] and plagiarism [4]. Since LLM-generated text can be indistinguishable from human-written text [5], it is necessary to develop effective methods for detecting such content.

Detection methods can generally be categorized into two paradigms: supervised and zero-shot approaches. Supervised methods typically train a classifier using both human-written and LLM-generated texts. However, such classifiers often fail to generalize well to texts generated by unseen LLMs [6, 7]. In contrast, zero-shot methods provide an alternative with more generalization ability and without the need for training [8]. These methods leverage various LLMs and exploit statistical classification metrics to assess how likely a text is to be generated by an LLM, such as the average log likelihood or the log-rank of each tokens within the detected text [9, 10, 5]. However, in real-world the source LLM of the detected text is often unknown, thus these zero-shot methods have to rely on a proxy LLM and suffer from a performance drop. This highlights the need to identify model-agnostic metrics for detection.

ReMoDetect [11] has recently proposed using the reward score—the output of a reward model that measures the extent to which a text aligns with human values—as a detection metric. The rationale is that powerful LLMs are typically trained to obtain high reward score, making this signal potentially model-agnostic. Specifically, ReMoDetect constructs a preference dataset from human and ChatGPT responses to the same prompt, and fine-tunes a pre-trained reward model for adaptation to the detection task. However, this fine-tuning inherits the drawbacks of supervised methods. As

---

[*]Corresponding author.

39th Conference on Neural Information Processing Systems (NeurIPS 2025).

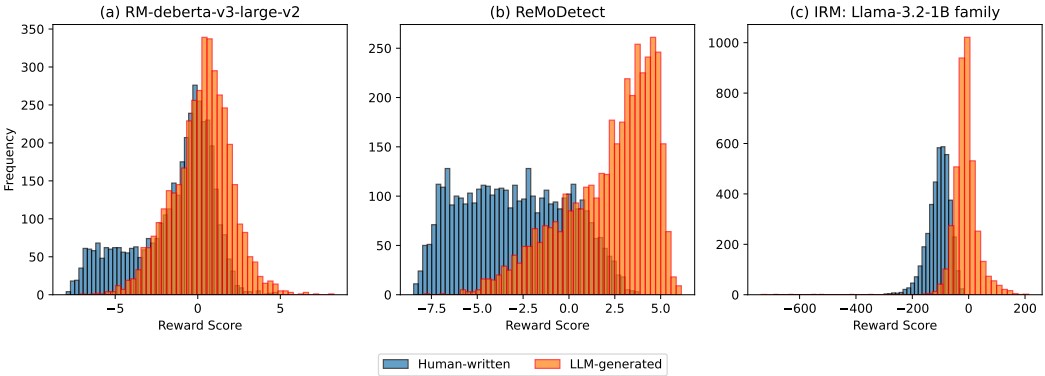

Figure 1: Distributions of reward scores for human-written and LLM-generated texts from the multi-LLM sub-task of the DetectRL benchmark. The source LLMs include GPT-3.5-turbo, PaLM-2-Bison, Claude-Instant, and LLaMA-2-70B. (a) shows distributions under RM-deberta-v3-large-v2, which is a pre-trained reward model. (b) shows distributions under ReMoDetect, which is initialized from RM-deberta-v3-large-v2 and fine-tuned on task-specific preference dataset. (c) illustrates distributions under an implicit reward model derived from Llama-3.2-1B family without additional training.

shown in Figure 1 (b), when ReMoDetect is evaluated on texts generated by unseen LLMs, there remains substantial overlap between the two score distributions of human-written and LLM-generated texts, suggesting the limited generalization ability.

In this paper, we propose IRM, a novel zero-shot method that leverages implicit reward models for LLM-generated text detection. Such implicit reward models can be derived from publicly available instruction-tuned and base models. Given a detected text $y$, an instruction-tuned model $\pi_\theta$ and a base model $\pi_{\mathrm{ref}}$, the implicit reward score of $y$ is formulated as $r(y) = \log \frac{\pi_\theta(y)}{\pi_{\mathrm{ref}}(y)}$. This score serves as the detection metric in our method, where a higher score indicates that the input text is more likely to be generated by LLMs. This formulation is built on Direct Preference Optimization (DPO) [12], which provides theoretical support for parameterizing implicit reward models. Compared to ReMoDetect, IRM requires neither preference collection nor additional training. As shown in Figure 1 (c), the two distributions under IRM are more separable than those in (b), suggesting that our zero-shot metric transfers well to unseen LLMs.

We evaluate IRM on the DetectRL benchmark [13] and demonstrate that IRM is an effective zero-shot method for LLM-generated text detection. Notably, with Llama-3.2-1B family, IRM achieves an average score of 91.77%, surpassing previous zero-shot baselines such as Log-Likelihood (77.46%) and Log-Rank (77.42%), as well as Binoculars (87.67%), a method which also utilizes two LLMs during detection. Furthermore, IRM even outperforms the supervised reward-based method, ReMoDetect (85.86%). These results underscore IRM's potential as a effective zero-shot detector for real-world applications.

## 2 Preliminaries

The mainstream alignment approach for LLMs is Reinforcement Learning from Human Feedback (RLHF) [14, 15], which relies on a reward model as a proxy for human preferences. In this section, we first introduce the concept of reward modeling, followed by an overview of the RLHF process.

**Reward modeling.** Typically, the reward model is initialized from a pre-trained language model with a additional linear head that outputs a scalar value. It is trained on a preference dataset, where human annotators compare multiple responses to the same prompt and rank them based on quality such as helpfulness or harmlessness. Formally, the preference is modeled using the Bradley-Terry (BT) model [16]:

$$p(y_1 \succ y_2 | x) = \frac{\exp(r(x, y_1))}{\exp(r(x, y_1)) + \exp(r(x, y_2))}, \tag{1}$$

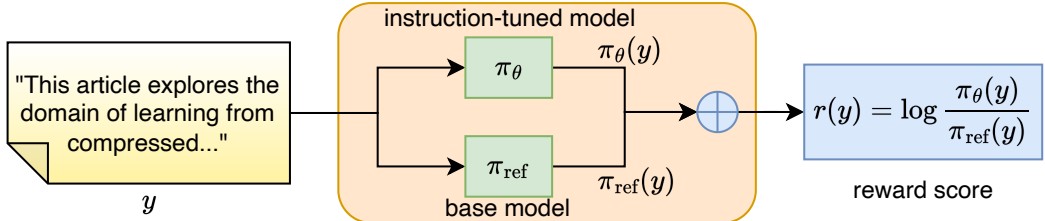

Figure 2: IRM leverages open-source instruction-tuned and base models to construct an implicit reward model. The resulting reward score is used as a detection metric, where a higher score indicates a higher probability that $y$ is generated by an LLM.

where $r$ denotes the reward model, $x$ is the prompt, and $y_1$, $y_2$ are two candidate responses. Given a prompt–response pair $(x, y)$, the reward model outputs a scalar score $r(x, y)$, where a higher score indicates stronger alignment with human values.

**RLHF.** During RLHF, the learned reward model is used to provide feedback for optimizing the language model. The LLM is framed as a policy model $\pi_\theta$, which generates responses conditioned on input prompts. The policy is optimized to maximize its expected reward while remaining close to a reference model $\pi_{\text{ref}}$, typically the original base LLM prior to alignment. This leads to the following optimization objective:

$$\max_{\pi_\theta} \mathbb{E}_{x \sim \mathcal{D}, y \sim \pi_\theta(y|x)}[r(x, y)] - \beta \mathbb{D}_{\text{KL}}[\pi_\theta(y|x) \| \pi_{\text{ref}}(y|x)], \tag{2}$$

where $x$ denotes the prompt, $y$ the response, $r$ the reward model, and $\beta$ a hyper-parameter controlling the strength of KL regularization.

## 3 Method

We introduce IRM, a zero-shot detection method for LLM-generated text detection. We formalize the approach and show how it can be instantiated using publicly available instruction-tuned and base models.

**Implicit reward model for detection.** Direct Preference Optimization (DPO) [12] offers a closed-form solution to the RLHF objective in Eq. 2:

$$\pi_\theta^*(y|x) = \frac{1}{Z(x)} \pi_{\text{ref}}(y|x) \exp\left(\frac{1}{\beta} r(x, y)\right), \tag{3}$$

where $Z(x) = \sum_y \pi_{\text{ref}}(y|x) \exp\left(\frac{1}{\beta} r(x, y)\right)$ is the partition function that normalizes the distribution. By rearranging this equation, we can express the reward as a function of the policy and reference models, yielding an implicit reward model:

$$r(x, y) = \beta \log \frac{\pi_\theta(y|x)}{\pi_{\text{ref}}(y|x)} + \beta \log Z(x). \tag{4}$$

For simplicity, we let $x = \varnothing$ and $y$ be the detected text. In this case, the term $\beta \log Z(x)$ becomes a constant across all detected texts. Importantly, applying a linear transformation to the reward scores does not affect commonly used classification metrics such as AUROC and $F_1$, as long as the threshold is adjusted accordingly. As a result, we define the detection metric as:

$$r(y) = \log \frac{\pi_\theta(y)}{\pi_{\text{ref}}(y)} = \sum_{i=1}^{L} \log \frac{\pi_\theta(y_i|y_{<i})}{\pi_{\text{ref}}(y_i|y_{<i})}, \tag{5}$$

where $L$ is the length of $y$ and $y_i$ denotes the $i$-th token. DPO also provides a theoretical justification for the effectiveness of the implicit reward model. As shown in Eq. 4, given a reward model $r(x, y)$

and a reference model $\pi_{\mathrm{ref}}$, one can analytically derive the optimal policy model $\pi_\theta^*$ under the RLHF objective. Conversely, given a learned policy model $\pi_\theta$ and a reference model $\pi_{\mathrm{ref}}$, one can recover a corresponding reward model such that $\pi_\theta$ is optimal with respect to that reward under the same reference model. This bidirectional relationship ensures that the implicit reward model is consistent with the behavior of the policy model relative to the reference model. Therefore, if the policy model aligns well with human preferences, the derived implicit reward model is also expected to exhibit strong performance in tasks such as LLM-generated text detection.

**Constructing IRM using instruction-tuned and base models.** Typically, publicly available LLMs release both instruction-tuned model and its corresponding base model. We leverage the instruction-tuned model as the policy model and the base model as the reference model. In this way, without any fine-tuning, we can obtain an implicit reward model for LLM-generated text detection. Given a detected text $y$, its reward score can be computed via Eq. 5. Moreover, the same formulation naturally extends to LLMs trained via iterative alignment. Suppose instruction model $\pi_T$ is obtained from the base model $\pi_0$ through $T$ rounds of optimization: $\pi_0 \rightarrow \pi_1 \rightarrow ... \rightarrow \pi_{T-1} \rightarrow \pi_T$. While the intermediate models $\pi_1, ..., \pi_{T-1}$ are typically not publicly released, we can still conceptually define the implicit reward at each step $i$ by applying Eq. 4:

$$r_i(x, y) = \beta \log \frac{\pi_{i+1}(y|x)}{\pi_i(y|x)} + \beta \log Z_i(x). \tag{6}$$

By summing the rewards over all iterations, we obtain the total reward:

$$\sum_{i=1}^{T} r_i(x, y) = \beta \log \frac{\pi_T(y|x)}{\pi_0(y|x)} + \beta \sum_{i=1}^{T} Z_i(x). \tag{7}$$

Since $x = \varnothing$, the term $\beta \sum_{i=1}^{T} Z_i(x)$ becomes a constant. As linear transformation does not affect the ranking of detected texts, the overall reward is determined by $\log \frac{\pi_T(y)}{\pi_0(y)}$. Therefore, given only the instruction-tuned and base models, we can construct an implicit reward model for detection without requiring access to intermediate checkpoints.

## 4 Experiments

### 4.1 Settings

**Datasets.** We conduct evaluations on a large benchmark, DetectRL [13], which covers various domains, multiple LLMs and diverse attacks scenarios in real-world settings. We follow the test setting of DetectRL in the evaluation of detection methods. The statistics of the benchmark are reported in Table 6 in Appendix A.1. Notably, the texts in DetectRL are mostly complete, unlike previous works where the texts are often truncated to prefixes of complete texts. This allows for a more realistic evaluation of detection methods, as texts are typically not truncated in real-world scenarios.

**Models.** We employ Gemma-2 [17], Llama-3.2, Gemma [18], Qwen-2 [19], and Qwen-2.5 [20] model families for implementation. These model families provide both publicly available base models and instruction-tuned models, which are essential for deriving the implicit reward model. Within these families, we focus on lightweight LLMs, such as Gemma-2-2B-it, and Llama-3.2-1B-Instruct, to ensure computational efficiency while maintaining competitive performance.

**Baselines.** We compare IRM with recent zero-shot methods, including Binoculars [21] and Lastde [22], as well as typical zero-shot baselines such as Log-Likelihood [9, 10], Rank [5], Log-Rank [5], LRR [23] and Fast-DetectGPT [24]. Since we employs reward score as detection metrics, we also include ReMoDetect [11] and its initialization, RM-Deberta-v3-large-v2, as well as reward models from RewardBench [25], such as GRM [26] as baselines. For practical consideration, we exclude methods that require additional LLMs to perturb the original text for metric computation, as the extra cost is too high in real-world settings.

**Metrics.** Follow the settings of DetectRL, we report AUROC and $F_1$ Score as the main evaluation metrics. AUROC is widely used for assessing zero-shot detection methods and $F_1$ Score provides a comprehensive evaluation of detector capabilities by balancing the Precision and Recall.

Table 1: The overall performance of zero-shot detection methods on the DetectRL benchmark, covering robustness and generalization tasks, which include various domains, LLMs and attacks scenarios. It also considers the effects of text length and real-world human writing characteristics. The reported average score is computed across all tasks. For each task, the best results are marked in **bold** and the second best results are marked by underline. More detailed detection results are available in Table 8, 9, 10 and 11 in Appendix A.2.

| Tasks Settings → | Multi-Domain | | Multi-LLM | | Multi-Attack | | Generalization | | | Length | | Human Writing | | Avg. |
| | | | | | | | Domain | LLM | Attack | Train | Test | | | |
| Methods ↓ | AUROC | $F_1$ | AUROC | $F_1$ | AUROC | $F_1$ | $F_1$ | $F_1$ | $F_1$ | $F_1$ | $F_1$ | AUROC | $F_1$ | |
| *Gemma-2-2B-it* | | | | | | | | | | | | | | |
| Log-Likelihood | 74.75 | 68.75 | 74.65 | 62.81 | 77.92 | 70.65 | 65.76 | 63.07 | 64.79 | 69.78 | 69.70 | 92.78 | 87.54 | 72.53 |
| Rank | 53.56 | 45.45 | 53.45 | 45.64 | 57.62 | 46.14 | 44.08 | 43.78 | 36.86 | 44.92 | 44.93 | 79.05 | 71.77 | 51.33 |
| Log-Rank | 75.78 | 69.81 | 75.62 | 65.36 | 78.75 | 71.37 | 66.79 | 64.85 | 66.30 | 69.50 | 68.79 | 93.02 | 88.05 | 73.38 |
| LRR | 77.43 | 70.81 | 76.81 | 66.62 | 79.28 | 71.32 | 68.20 | 66.05 | 68.30 | 65.75 | 63.44 | 91.41 | 85.75 | 73.17 |
| Fast-DetectGPT | 70.26 | 63.62 | 69.51 | 59.85 | 73.30 | 66.02 | 62.42 | 59.94 | 62.63 | 56.60 | 61.66 | 89.56 | 83.61 | 67.61 |
| Lastde | 60.62 | 55.83 | 59.70 | 54.91 | 66.11 | 56.31 | 54.45 | 53.61 | 45.29 | 47.25 | 53.76 | 90.64 | 85.63 | 60.32 |
| *Gemma-2-2B family (Gemma-2-2B-it+Gemma-2-2B)* | | | | | | | | | | | | | | |
| Binoculars | 80.85 | 74.57 | 80.43 | 71.24 | 83.00 | 76.58 | 73.11 | 70.95 | 73.64 | **73.03** | **71.71** | **93.29** | **89.07** | **77.81** |
| IRM | **84.65** | **77.70** | **86.56** | **77.43** | **88.38** | **79.02** | **76.97** | **76.62** | **78.64** | 60.30 | 58.15 | 81.19 | 76.04 | 77.05 |
| *Llama-3.2-1B-Instruct* | | | | | | | | | | | | | | |
| Log-Likelihood | 79.40 | 73.49 | 79.67 | 71.34 | 82.65 | 76.31 | 72.29 | 70.84 | 73.32 | 72.87 | 70.98 | 94.02 | 89.85 | 77.46 |
| Rank | 59.65 | 53.70 | 59.50 | 50.84 | 63.60 | 52.35 | 51.02 | 49.52 | 41.91 | 51.20 | 49.12 | 84.37 | 78.20 | 57.31 |
| Log-Rank | 79.64 | 73.41 | 79.78 | 71.50 | 82.55 | 75.37 | 72.05 | 70.54 | 73.32 | 73.08 | 71.46 | 93.94 | 89.81 | 77.42 |
| LRR | 77.99 | 70.66 | 77.79 | 69.33 | 79.56 | 70.65 | 69.25 | 68.28 | 68.38 | 69.97 | 68.05 | 91.62 | 86.72 | 74.48 |
| Fast-DetectGPT | 68.04 | 58.44 | 68.62 | 58.72 | 69.58 | 62.47 | 56.76 | 58.40 | 60.90 | 45.94 | 55.84 | 60.83 | 53.12 | 59.82 |
| Lastde | 65.11 | 56.03 | 59.70 | 54.91 | 69.59 | 69.42 | 53.40 | 53.14 | 59.47 | 41.59 | 49.79 | 87.52 | 82.00 | 61.67 |
| *Llama-3.2-1B family (Llama-3.2-1B-Instruct+Llama-3.2-1B)* | | | | | | | | | | | | | | |
| Binoculars | 92.48 | 85.63 | 92.08 | 86.67 | 93.04 | 87.02 | 85.07 | 83.18 | 85.60 | 80.62 | 81.88 | **94.63** | **91.82** | 87.67 |
| IRM | **97.97** | **93.75** | **97.24** | **92.12** | **97.19** | **92.01** | **90.23** | **90.72** | **90.87** | **82.34** | **83.34** | 94.48 | 90.78 | **91.77** |

**Implementation Details.** All experiments are conducted on two NVIDIA RTX 4090 GPUs (24GB each). All model and datasets used in this paper are fully detailed and referenced in Table 7 in Appendix A.1.

## 4.2 Main Results

### 4.2.1 Zero-shot LLM-generated Text Detection

In Table 1, we present the overall performance of IRM and other zero-shot detection methods on the DetectRL benchmark. Detailed results for each sub-task are reported in Table 8, 9, 10 and 11 in Appendix A.2. With the Llama-3.2-1B family, IRM achieves the highest average score, improving the best result by 4.1%, from 87.67% to 91.77%. Using the Gemma-2-2B family, IRM also attains a competitive average score of 77.05%, slightly below the best result of 77.81%. All methods, except Fast-DetectGPT, perform better when using the Llama-3.2-1B family than with the Gemma-2-2B family, indicating that Llama-3.2-1B family is more suitable for these zero-shot detection approaches.

### 4.2.2 Reward-based LLM-generated Text Detection

In Table 2, we present the overall performance of IRM and other reward models on the DetectRL benchmark. With the Llama-3.2-1B family, IRM achieves the highest average score, improving the best result by 5.8%, from 85.86% to 91.77%. Notably, IRM outperforms ReMoDetect, a supervised reward model that fine-tunes a pre-trained reward model (RM-deberta-v3-large-v2) using task-specific data. In addition, IRM also outperforms competitive reward models from the RewardBench benchmark, such as GRM-Llama3.2-3B and GRM-gemma2-2B, when evaluated on DetectRL benchmark. This suggests a potential mismatch between reward models optimized for preference classification and the requirements of LLM-generated text detection, indicating that directly leveraging such models for detection may be suboptimal.

## 4.3 Analysis

**Robustness to domains.** As shown in Table 8, IRM outperforms other detection methods on 6 out of 8 evaluation metrics and achieves the best average performance across all domains. While its performance varies slightly across domains—for instance, achieving particularly strong results on the arXiv dataset and relatively lower scores on the Review dataset—it maintains overall superiority throughout. For example, using the Llama-3.2-1B family, IRM achieves an AUROC of 98.56% and

Table 2: The overall performance of reward-based detection methods on the DetectRL benchmark, covering robustness and generalization tasks, which include various domains, LLMs and attacks scenarios. It also considers the effects of text length and real-world human writing characteristics. The reported average score is computed across all tasks. For each task, the best results are marked in **bold** and the second best results are marked by underline. More detailed detection results are available in Table 8, 9, 10 and 11 in Appendix A.2.

| Tasks Settings → | Multi-Domain | | Multi-LLM | | Multi-Attack | | Generalization | | | Length | | Human Writing | | Avg. |
|---|---|---|---|---|---|---|---|---|---|---|---|---|---|---|
| | | | | | | | Domain | LLM | Attack | Train | Test | | | |
| Reward Models ↓ | AUROC | $F_1$ | AUROC | $F_1$ | AUROC | $F_1$ | $F_1$ | $F_1$ | $F_1$ | $F_1$ | $F_1$ | AUROC | $F_1$ | |
| ReMoDetect | 91.99 | 85.95 | 90.41 | 83.17 | 91.22 | 82.09 | 79.08 | 78.36 | 80.22 | 88.53 | 85.87 | 91.63 | 87.65 | 85.86 |
| RM-Deberta-v3-large-v2 | 73.08 | 65.01 | 70.37 | 64.91 | 71.34 | 63.44 | 56.70 | 62.28 | 61.21 | 79.68 | 72.71 | 81.48 | 74.62 | 68.99 |
| GRM-gemma2-2B | 51.07 | 31.20 | 50.57 | 45.32 | 50.69 | 59.97 | 27.53 | 41.58 | 56.30 | 45.71 | 53.97 | 63.69 | 60.37 | 49.07 |
| GRM-Llama3.2-3B | 45.47 | 21.27 | 46.15 | 22.85 | 48.56 | 30.80 | 19.12 | 19.63 | 29.15 | 46.81 | 49.99 | 60.71 | 54.64 | 38.09 |
| *IRM* | | | | | | | | | | | | | | |
| Gemma-2-2B family | 84.65 | 77.70 | 86.56 | 77.43 | 88.38 | 79.02 | 76.97 | 76.62 | 78.64 | 60.30 | 58.15 | 81.19 | 76.04 | 77.05 |
| Llama-3.2-1B family | **97.97** | **93.75** | **97.24** | **92.12** | **97.19** | **92.01** | **90.23** | **90.72** | **90.87** | 82.34 | 83.34 | **94.48** | **90.78** | **91.77** |

Table 3: The performance of IRM using the Llama-3.2-1B family on various attacks to human-written texts.

| DIPPER Paraphrase | | | | Polish using LLMs | | | | Back Translation | | | |
|---|---|---|---|---|---|---|---|---|---|---|---|
| Pre | Rec | $F_1$ | AUROC | Pre | Rec | $F_1$ | AUROC | Pre | Rec | $F_1$ | AUROC |
| 93.02 | 83.33 | 87.91 | 94.39 | 77.76 | 79.76 | 78.75 | 85.36 | 93.11 | 87.10 | 90.01 | 95.54 |

an $F_1$ Score of 95.02% on the arXiv dataset, and an AUROC of 97.03% and an $F_1$ Score of 91.65% on the Review dataset. One possible explanation is that positive reviews are tend to be more neutral or benign compared to negative reviews, making them more likely to receive higher reward scores. This subtle property of the data may slightly reduce the separation margin between human-written and LLM-generated texts. Still, IRM maintains strong performance overall, underscoring its effectiveness across diverse domains.

**Robustness to LLMs.** As shown in Table 9, IRM outperforms other detection methods on 7 out of 8 evaluation metrics and achieves the best average performance across all LLMs. While IRM demonstrates strong generalization across a wide range of generation models, its performance can be further enhanced when the instruction-tuned and base models used to construct the implicit reward model share a similar model-family background with the generation model. For example, when using the Gemma-2-2B family, IRM performs particularly well in detecting texts generated by PaLM-2-bison, achieving an AUROC of 88.65% and an $F_1$ Score of 79.41%. Similarly, with the Llama-3.2-1B family, IRM demonstrates outstanding performance in detecting Llama-2-70B's outputs, achieving an AUROC of 99.35% and an $F_1$ Score of 96.58%.

**Robustness to attacks.** As shown in Table 10, IRM outperforms other detection methods on 8 out of 10 evaluation metrics and achieves the highest average performance across all attacks applied to LLM-generated texts. For example, when using Llama-3.2-1B family, IRM achieves an average AUROC of 97.19% and an average $F_1$ Score of 92.01%. While IRM demonstrates strong robustness against attacks on LLM-generated texts, its performance is relatively lower but still competitive when attacks are applied to human-written texts, as shown in Table 11. For example, with the Llama-3.2-1B family, IRM achieves an average AUROC of 94.48% and an average $F_1$ Score of 90.78%, which are slightly below than the best results: 94.63% for AUROC and 91.82% for $F_1$ Score. We further investigate which attack type has the greatest impact on IRM's performance. As shown in Table 3, with the Llama-3.2-1B family, IRM is particularly affected by the polishing attack, where LLMs are prompted to improve human-written texts. This result is intuitive: since the polished texts are generated by LLMs and typically exhibit improved fluency or coherence, they tend to receive higher reward scores than their original, unpolished counterparts. As a result, the distinction between human-written and LLM-generated texts become less clear, leading to a moderate drop in detection performance for IRM.

**Generalization of IRM.** In the generalization task, following the DetectRL benchmark settings, we evaluate each sub-task using the optimal threshold from other sub-tasks and calculate the average $F_1$ Score across all sub-tasks. As shown in Table 1, IRM exhibits strong generalization capability

Table 4: The performance comparison of baseline zero-shot methods using instruction-tuned and base models. Gemma-2-2B-it and Llama-3.2-1B-Instruct are instruction-tuned models, while Gemma-2-2B and Llama-3.2-1B are their corresponding base models.

| Model | Log-Likelihood | Rank | Log-Rank | LRR | Fast-DetectGPT | Lastde |
|---|---|---|---|---|---|---|
| Llama-3.2-1B-Instruct | 77.46 | 57.31 | 77.42 | 74.48 | 59.82 | 61.67 |
| Llama-3.2-1B | 66.44 | 50.80 | 66.81 | 65.78 | 63.80 | 59.13 |
| $\Delta$ | -11.02 | -6.51 | -10.61 | -8.70 | +3.98 | -2.54 |
| Gemma-2-2B-it | 72.53 | 51.33 | 73.38 | 73.17 | 67.61 | 60.32 |
| Gemma-2-2B | 65.74 | 47.23 | 67.24 | 69.55 | 63.08 | 59.25 |
| $\Delta$ | -6.79 | -4.10 | -6.14 | -3.62 | -4.53 | -1.07 |

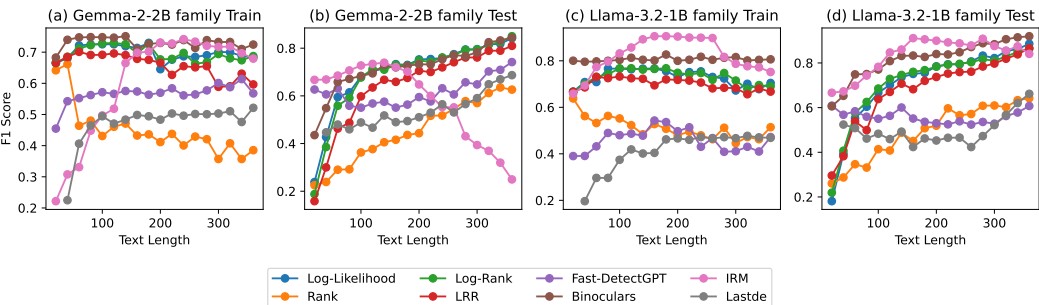

Figure 3: The performance of various zero-shot detection methods across different text lengths during training-time and test-time.

across different domains, LLMs and attacks. For instance, when using Llama-3.2-1B family, IRM achieves $F_1$ Score of 90.23%, 90.72% and 90.87% on domain, LLM and attack generalization tasks. These results suggest that the decision boundary of the implicit reward score is stable across various conditions, highlighting IRM's superior generalization ability.

**Effectiveness of instruction-tuned models for detection methods.** As prior works often utilize base models when implementing detection methods, we follow this practice and implement baseline methods using both Llama-3.2-1B and Gemma-2-2B. As shown in Figure 4, using instruction-tuned models consistently improves the performance of 5 out of 6 baseline methods. For instance, when using Llama-3.2-1B-Instruct, the performance of Log-Likelihood improves from 66.44% to 77.46%. Similarly, with Gemma-2-2B-it, the performance increases from 65.74% to 72.53%. These results suggest that instruction-tuned models are generally more effective for implementing detection methods.

**Impact of text length.** In the length task, following the DetectRL benchmark settings, we analyze the impact of text length on IRM's performance. For the train sub-task, we use the optimal threshold derived from the dataset with a length interval of 160-180 and apply it to datasets with other length intervals. Conversely, for the test sub-task, the optimal threshold of each dataset is evaluated on the dataset with a 160-180 length interval. We refer to the dataset used to derive the threshold as the training dataset, and the dataset used to evaluate the threshold as the test dataset. As shown in Table 1, when using Llama-3.2-1B family, IRM achieves the highest $F_1$ Score of 82.34% and 83.34% on the train and test sub-tasks, respectively. However, when using Gemma-2-2B family, IRM's performance drops significantly compared to other methods. To further investigate the influence of text length, we present the detailed results of train and test sub-tasks in Figure 3. IRM exhibits a distinct performance pattern trend compared to the other methods. Specifically, IRM tends to perform better than the length interval of the training dataset is similar to that of the test dataset. For example, using Gemma-2-2B family, thresholds derived from texts with lengths of 180-260 words perform better on 160-180-word texts (Figure 3 (a)), while the threshold from texts with 160-180-word is more effective for texts of 80-140 words (Figure 3 (b)). Similarly, using Llama-3.2-1B family, thresholds derived from texts of 140-260 words perform better on 160-180-word texts (Figure 3 (c)), whereas the threshold from 160-180-word texts performs better on texts of 140-200 words (Figure 3 (d)). This phenomenon may

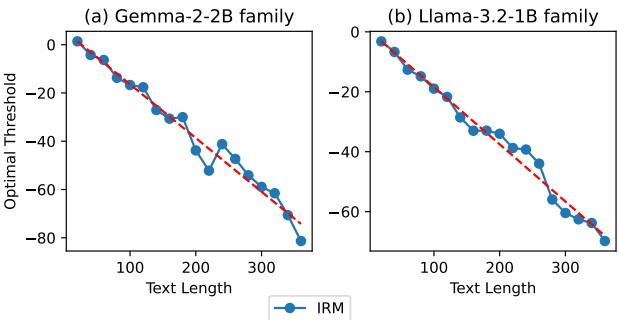

Figure 4: Optimal classification thresholds of IRM across datasets with varying text length intervals. The solid line shows the threshold values for each length interval (from 0 to 360 words), and the red dashed line represents a linear fit to illustrate the overall trend.

Table 5: The performance of IRM with different model versions and sizes.

| Tasks Settings → | Multi-Domain | | Multi-LLM | | Multi-Attack | | Generalization | | | Length | | Human Writing | | Avg. |
|---|---|---|---|---|---|---|---|---|---|---|---|---|---|---|
| Model Family ↓ | AUROC | $F_1$ | AUROC | $F_1$ | AUROC | $F_1$ | Domain $F_1$ | LLM $F_1$ | Attack $F_1$ | Train $F_1$ | Test $F_1$ | AUROC | $F_1$ | |
| Qwen2-0.5B | 90.14 | 83.57 | 89.43 | 82.54 | 90.47 | 83.55 | 82.48 | 81.59 | 83.34 | 74.17 | 70.67 | 86.11 | 78.27 | 82.79 |
| Qwen2-1.5B | 89.68 | 82.49 | 89.64 | 82.59 | 89.44 | 81.59 | 81.80 | 81.82 | 81.33 | 76.21 | 74.52 | 85.22 | 77.03 | 82.57 |
| Qwen2.5-0.5B | 79.44 | 76.28 | 77.84 | 74.02 | 78.48 | 73.55 | 74.75 | 73.35 | 73.03 | 66.80 | 62.30 | 74.18 | 68.10 | 73.24 |
| Qwen2.5-1.5B | 76.22 | 73.51 | 74.75 | 69.68 | 74.70 | 68.93 | 69.11 | 68.56 | 68.33 | 65.54 | 63.17 | 70.55 | 62.86 | 69.69 |
| Gemma-2B | 91.84 | 84.06 | 92.08 | 84.77 | 93.17 | 85.48 | 83.19 | 82.18 | 85.13 | 58.06 | 68.00 | 92.50 | 87.90 | 83.72 |
| Gemma-2-2B | 84.65 | 77.70 | 86.56 | 77.43 | 88.38 | 79.02 | 76.97 | 76.62 | 78.64 | 60.30 | 58.15 | 81.19 | 76.04 | 77.05 |
| Llama-3.2-1B | 97.97 | 93.75 | 97.24 | 92.12 | 97.19 | 92.01 | 90.23 | 90.72 | 90.87 | 82.34 | 83.34 | 94.48 | 90.78 | 91.77 |
| Llama-3.2-3B | 97.31 | 92.51 | 97.25 | 92.15 | 97.21 | 91.69 | 91.68 | 90.67 | 91.37 | 78.59 | 80.97 | 93.85 | 88.85 | 91.08 |

be attributed to the way the implicit reward score is computed. As shown in Eq. 5, the reward is accumulated over all tokens, so the total reward varies with text length. Consequently, the reward distribution shifts across different length intervals, which affects the optimal threshold value. This trend is illustrated in Figure 4, where the optimal threshold of IRM changes with text length, showing an approximately linear relationship. Notably, for the Gemma-2-2B family, the optimal threshold fluctuates and deviates from the fitted red dashed line within the 160–220-word intervals. Besides, we can also observe that, with Gemma-2-2B family, the optimal threshold of IRM fluctuates and deviates from the red dash within 160-220-words intervals, which can explain the performance drop of IRM when using Gemma-2-2B family. Since this length range corresponds to the train and test dataset in the length task, such instability in threshold values may explains the observed performance drop of IRM when using the Gemma-2-2B family in this task.

**Impact of model version and size.** We further investigate the impact of model version and size on the performance of IRM by implementing it across various model families, as presented in Table 5. When using the Llama-3.2-1B family, IRM achieves the best average score of 91.77%, slightly higher than the 91.08% achieved with the larger Llama-3.2-3B family. A similar trend is observed in the Qwen2 and Qwen2.5 families: for example, IRM attains 73.24% with Qwen2.5-0.5B family, outperforming Qwen2.5-1.5B family (69.69%). Besides, IRM achieves 83.72% with the Gemma-2B family, compared to 77.05% with Gemma-2-2B family. These results suggest that larger or newer versions of a model do not always lead to better detection performance.

## 5 Related Works

**LLM-generated text detection.** Since generative language models have shown remarkable capability in text generation, detecting AI-generated texts has attracted significant research attention. Current research primarily focus on supervised methods and zero-shot methods for LLMs. Supervised methods involve collecting labeled data and training classifiers for distinguishing human-written and machine-generated texts [27–29, 11]. However, it has been shown that many supervised methods often suffer from overfitting to the training set [6, 7], limiting their generalization ability. In contrast, zero-shot methods have shown better generalization ability, making them a promising alternative. Current zero-shot methods are based on pretrained LLMs and utilize heuristic metrics to classify

LLM-generated texts and human-written texts. These metrics include likelihood [9, 10], entropy, rank, log-rank [5], probability curvature [8, 24, 30], etc. Additionally, perturbing texts and measuring the variation of metrics is a commonly used augmentation method for constructing better metrics [22, 31–33], but it incurs extra computational costs. Our method is a zero-shot method that leverages the characteristics of aligned LLMs and derives a metric from LLMs in an off-the-shelf manner, without the need of additional training. Additionally, our method does not require perturbing texts to compute the variation of metrics, avoiding extra computational costs.

**Implicit reward model.**    In LLM alignment, the primary focus is on obtaining a well-aligned LLM, rather than a high-quality reward model. Thus, although DPO and other methods use implicit reward models to implement LLM alignment, the implicit reward models obtained after training are typically not reused. Recently, some research has leveraged implicit reward models to label process rewards for reasoning tasks [34]. Since our method uses the reward score as a metric for distinguishing human-written and LLM-generated texts, we propose leveraging implicit reward models to compute the reward score for texts.

## 6    Discussion and Conclusion

In this paper, we propose IRM, a novel method for zero-shot detection of LLM-generated text. IRM utilizes implicit reward models to compute reward scores as detection metrics. Experiments on the DetectRL benchmark demonstrate that IRM is an effective zero-shot method for LLM-generated text detection. Notably, with Llama-3.2-1B family, IRM achieves an average score of 91.77%, surpassing previous zero-shot baselines such as Log-Likelihood, Log-Rank and Binoculars. Furthermore, IRM even outperforms the supervised baseline ReMoDetect, showing that IRM remains highly effective without any task-specific adaptation. These results underscore IRM's potential as a effective zero-shot detector for real-world applications.

**Limitations and future works.**    In our experiments, we focus exclusively on lightweight LLMs, excluding larger LLMs due to computational constraints. Similarly, we do not consider generative reward models as baselines, as they typically require larger LLMs and more computation resources. Besides, we restrict our analysis to deriving IRM from LLMs within the same model family. For cases where only instruction-tuned models are available, deriving corresponding IRM is not feasible. We leave how to effectively leverage LLMs from different model families for future work.

**Societal impact.**    IRM contributes to the responsible use of LLMs by providing a zero-shot method for detecting LLM-generated text. This capability can enhance transparency and help mitigate misuse in high-stakes domains such as education and media. On the other hand, misclassifications may inadvertently suppress human-generated text, highlighting that need for continuous development more powerful detection methods.

## Acknowledgments and Disclosure of Funding

We thank all the anonymous reviewers for their insightful and valuable feedback on this paper. This work is supported by the National Natural Science Foundation of China (Grant No.U21B2009).

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

# A  Details of Main Experiments

## A.1  Details of Models and Datasets

Table 6: Statistic of DetectRL benchmark.

| Setting | Sub Setting | Size |
|---|---|---|
| Multi-Domain | Academic | 2008 |
| | News | 2008 |
| | Creative | 2008 |
| | Social Media | 2008 |
| Multi-LLM | GPT-3.5-turbo | 2008 |
| | Claude-instant | 2008 |
| | PaLM-2-bison | 2008 |
| | Llama-2-70b | 2008 |
| Multi-Attack | Direct | 2016 |
| | Prompt | 2032 |
| | Paraphrase | 2016 |
| | Perturbation | 2016 |
| | Data Mixing | 2008 |
| Varing Text Length | - | 16200 |
| Human Writing | Direct | 2016 |
| | Paraphrase | 2016 |
| | Perturbation | 2016 |
| | Data Mixing | 2012 |

Table 7: Dataset and model details.

| Name Used | Full Name | Author | Source |
|---|---|---|---|
| DetectRL | DetectRL | [13] | GitHub |
| Llama-3.2-3B-Instruct | meta-llama/Llama-3.2-3B-Instruct | meta | HuggingFace |
| Llama-3.2-3B | meta-llama/Llama-3.2-3B | meta | HuggingFace |
| Llama-3.2-1B-Instruct | meta-llama/Llama-3.2-1B-Instruct | meta | HuggingFace |
| Llama-3.2-1B | meta-llama/Llama-3.2-1B | meta | HuggingFace |
| Gemma-2-2B-it | google/gemma-2-2b-it | [17] | HuggingFace |
| Gemma-2-2B | google/gemma-2-2b | [17] | HuggingFace |
| Gemma-2B-it | google/gemma-2-2b-it | [18] | HuggingFace |
| Gemma-2B | google/gemma-2-2b | [18] | HuggingFace |
| Qwen2.5-1.5B-Instruct | Qwen/Qwen2.5-1.5B-Instruct | [20] | HuggingFace |
| Qwen2.5-1.5B | Qwen/Qwen2.5-1.5B | [20] | HuggingFace |
| Qwen2.5-0.5B-Instruct | Qwen/Qwen2.5-0.5B-Instruct | [20] | HuggingFace |
| Qwen2.5-0.5B | Qwen/Qwen2.5-0.5B | [20] | HuggingFace |
| Qwen2-1.5B-Instruct | Qwen/Qwen2-1.5B-Instruct | [19] | HuggingFace |
| Qwen2-1.5B | Qwen/Qwen2-1.5B | [19] | HuggingFace |
| Qwen2-0.5B-Instruct | Qwen/Qwen2-0.5B-Instruct | [19] | HuggingFace |
| Qwen2-0.5B | Qwen/Qwen2-0.5B | [19] | HuggingFace |
| RM-Deberta-v3-large-v2 | OpenAssistant/reward-model-deberta-v3-large-v2 | OpenAssistant | HuggingFace |
| ReMoDetect | hyunseoki/ReMoDetect-deberta | [11] | HuggingFace |
| GRM-gemma2-2B | Ray2333/GRM-gemma2-2B-rewardmodel-ft | [26] | HuggingFace |
| GRM-Llama3.2-3B | Ray2333/GRM-Llama3.2-3B-rewardmodel-ft | [26] | HuggingFace |

## A.2 Detailed Results of Main Results

Table 8: The performance of various detection methods in the multi-domain task. The best results are marked in **bold** and the second best results are marked by underline.

| Metrics → | AUROC | $F_1$ | AUROC | $F_1$ | AUROC | $F_1$ | AUROC | $F_1$ | AUROC | $F_1$ |
|---|---|---|---|---|---|---|---|---|---|---|
| **Domain Settings →** | ArXiv | | XSum | | Writing | | Review | | Avg. | |
| *Gemma-2-2B-it* | | | | | | | | | | |
| Log-Likelihood | 74.48 | 68.01 | 59.84 | 54.29 | 81.23 | 74.37 | 83.44 | 78.34 | 74.75 | 68.75 |
| Rank | 50.87 | 41.75 | 41.25 | 28.91 | 60.26 | 54.23 | 61.86 | 56.92 | 53.56 | 45.45 |
| Log-Rank | 76.11 | 70.78 | 61.23 | 55.84 | 81.18 | 74.43 | 84.60 | 78.19 | 75.78 | 69.81 |
| LRR | 78.98 | 73.43 | 65.07 | 60.53 | 78.80 | 70.29 | 86.86 | 79.01 | 77.43 | 70.81 |
| Fast-DetectGPT | 73.25 | 67.17 | 70.92 | 65.08 | 68.41 | 61.06 | 68.49 | 61.18 | 70.26 | 63.62 |
| Lastde | 65.63 | 58.99 | 58.50 | 52.64 | 58.52 | 52.81 | 59.84 | 58.89 | 60.62 | 55.83 |
| *Gemma-2-2B family (Gemma-2-2B-it+Gemma-2-2B)* | | | | | | | | | | |
| Binoculars | 84.07 | 79.69 | 69.40 | 64.60 | 84.60 | 75.80 | 85.34 | 78.20 | 80.85 | 74.57 |
| IRM | 97.17 | 91.24 | 96.55 | 90.80 | 76.47 | 69.18 | 68.42 | 59.60 | 84.65 | 77.70 |
| *Llama-3.2-1B-Instruct* | | | | | | | | | | |
| Log-Likelihood | 79.00 | 72.61 | 67.10 | 61.74 | 83.15 | 76.22 | 88.37 | 83.38 | 79.40 | 73.49 |
| Rank | 58.48 | 51.12 | 44.80 | 37.06 | 59.81 | 56.76 | 75.50 | 69.87 | 59.65 | 53.70 |
| Log-Rank | 79.36 | 73.63 | 67.55 | 60.60 | 83.12 | 76.54 | 88.56 | 82.89 | 79.64 | 73.41 |
| LRR | 77.46 | 69.80 | 65.95 | 59.30 | 81.25 | 74.68 | 87.29 | 78.85 | 77.99 | 70.66 |
| Fast-DetectGPT | 85.71 | 78.26 | 78.71 | 72.16 | 42.42 | 24.82 | 65.33 | 58.50 | 68.04 | 58.44 |
| Lastde | 80.17 | 71.95 | 65.55 | 59.82 | 52.15 | 38.66 | 62.59 | 53.68 | 65.11 | 56.03 |
| *Llama-3.2-1B family (Llama-3.2-1B-Instruct+Llama-3.2-1B)* | | | | | | | | | | |
| Binoculars | 93.17 | 86.51 | 88.43 | 81.16 | 92.03 | 84.95 | 96.27 | 89.91 | 92.48 | 85.63 |
| IRM | **98.56** | **95.02** | **98.45** | **95.15** | 97.84 | 93.19 | **97.03** | **91.65** | **97.97** | **93.75** |
| *Reward Models* | | | | | | | | | | |
| ReMoDetect | 92.34 | 84.63 | 80.74 | 75.92 | **98.12** | **93.33** | 96.75 | 89.92 | 91.99 | 85.95 |
| RM-Deberta-v3-large-v2 | 58.16 | 45.78 | 59.37 | 52.06 | 95.49 | 89.59 | 79.30 | 72.62 | 73.08 | 65.01 |
| GRM-gemma2-2B | 33.30 | 1.18 | 63.47 | 57.68 | 70.16 | 64.95 | 37.33 | 0.98 | 51.07 | 31.20 |
| GRM-Llama3.2-3B | 45.94 | 19.45 | 43.20 | 3.49 | 66.08 | 58.82 | 26.66 | 3.31 | 45.47 | 21.27 |

Table 9: The performance of various detection methods in the multi-LLM task. The best results are marked in **bold** and the second best results are marked by underline.

| Metrics → | AUROC | $F_1$ | AUROC | $F_1$ | AUROC | $F_1$ | AUROC | $F_1$ | AUROC | $F_1$ |
|---|---|---|---|---|---|---|---|---|---|---|
| LLM Settings → | GPT-3.5 | | Claude | | PaLM-2 | | Llama-2 | | Avg. | |
| *Gemma-2-2B-it* | | | | | | | | | | |
| Log-Likelihood | 78.20 | 69.58 | 51.97 | 29.87 | 80.40 | 73.01 | 88.03 | 78.78 | 74.65 | 62.81 |
| Rank | 55.17 | 47.92 | 48.46 | 41.12 | 52.18 | 44.44 | 57.99 | 49.09 | 53.45 | 45.64 |
| Log-Rank | 78.48 | 69.46 | 54.43 | 38.13 | 80.76 | 73.14 | 88.81 | 80.73 | 75.62 | 65.36 |
| LRR | 76.60 | 68.24 | 62.97 | 48.44 | 79.08 | 69.57 | 88.58 | 80.25 | 76.81 | 66.62 |
| Fast-DetectGPT | 67.94 | 62.45 | 38.21 | 20.61 | 81.85 | 74.71 | 90.02 | 81.62 | 69.51 | 59.85 |
| Lastde | 57.40 | 58.63 | 44.49 | 29.57 | 65.51 | 63.65 | 71.39 | 67.78 | 59.70 | 54.91 |
| *Gemma-2-2B family (Gemma-2-2B-it+Gemma-2-2B)* | | | | | | | | | | |
| Binoculars | 77.48 | 69.55 | 62.21 | 48.61 | 89.44 | 82.36 | 92.61 | 84.43 | 80.43 | 71.24 |
| IRM | 84.03 | 75.68 | 86.46 | 78.44 | 88.65 | 79.41 | 87.11 | 76.21 | 86.56 | 77.43 |
| *Llama-3.2-1B-Instruct* | | | | | | | | | | |
| Log-Likelihood | 83.26 | 77.22 | 59.05 | 45.91 | 83.76 | 76.20 | 92.63 | 86.03 | 79.67 | 71.34 |
| Rank | 63.72 | 55.82 | 51.31 | 36.78 | 58.10 | 51.88 | 64.86 | 58.90 | 59.50 | 50.84 |
| Log-Rank | 82.98 | 74.79 | 59.73 | 48.50 | 83.75 | 76.31 | 92.67 | 86.41 | 79.78 | 71.50 |
| LRR | 79.73 | 71.92 | 60.56 | 49.61 | 80.75 | 72.97 | 90.13 | 82.81 | 77.79 | 69.33 |
| Fast-DetectGPT | 66.27 | 61.23 | 44.10 | 23.85 | 79.15 | 72.33 | 84.96 | 77.47 | 68.62 | 58.72 |
| Lastde | 57.91 | 49.93 | 54.02 | 42.85 | 67.54 | 60.69 | 76.18 | 68.69 | 63.91 | 55.54 |
| *Llama-3.2-1B family (Llama-3.2-1B-Instruct+Llama-3.2-1B)* | | | | | | | | | | |
| Binoculars | 92.26 | 84.89 | 81.38 | 74.40 | 95.56 | **90.90** | 99.10 | 96.50 | 92.08 | 86.67 |
| IRM | **97.40** | **92.22** | **96.27** | **89.79** | **95.95** | 89.88 | **99.35** | **96.58** | **97.24** | **92.12** |
| *Reward Models* | | | | | | | | | | |
| ReMoDetect | 96.14 | 89.24 | 82.26 | 76.74 | 87.82 | 79.01 | 95.41 | 87.68 | 90.41 | 83.17 |
| RM-Deberta-v3-large-v2 | 78.16 | 70.24 | 65.27 | 51.88 | 64.61 | 71.44 | 73.45 | 66.08 | 70.37 | 64.91 |
| GRM-gemma2-2B | 60.93 | 63.53 | 53.03 | 56.21 | 41.78 | 1.75 | 46.52 | 59.78 | 50.57 | 45.32 |
| GRM-Llama3.2-3B | 59.63 | 46.36 | 36.70 | 0.98 | 43.56 | 25.71 | 44.69 | 18.33 | 46.15 | 22.85 |

Table 10: The performance of various detection methods in the multi-attack task. The best results are marked in **bold** and the second best results are marked by underline.

| Metrics → | AUROC | $F_1$ | AUROC | $F_1$ | AUROC | $F_1$ | AUROC | $F_1$ | AUROC | $F_1$ | AUROC | $F_1$ |
|---|---|---|---|---|---|---|---|---|---|---|---|---|
| Attack Settings → | Direct | | Prompt | | Paraph. | | Perturb | | Mixing | | Avg. | |
| *Gemma-2-2B-it* | | | | | | | | | | | | |
| Log-Likelihood | 94.95 | 88.15 | 90.97 | 84.30 | 70.95 | 63.77 | 59.70 | 54.04 | 73.01 | 62.96 | 77.92 | 70.65 |
| Rank | 79.26 | 69.84 | 79.06 | 70.58 | 67.04 | 54.58 | 14.08 | 00.00 | 48.64 | 35.71 | 57.62 | 46.14 |
| Log-Rank | 95.25 | 88.56 | 91.21 | 84.38 | 71.40 | 64.61 | 61.72 | 55.57 | 74.19 | 63.71 | 78.75 | 71.37 |
| LRR | 93.16 | 85.08 | 88.92 | 80.63 | 70.88 | 63.38 | 67.39 | 59.72 | 76.07 | 67.77 | 79.28 | 71.32 |
| Fast-DetectGPT | 90.59 | 83.81 | 84.84 | 81.34 | 73.33 | 68.05 | 48.55 | 35.48 | 69.16 | 61.41 | 73.30 | 66.02 |
| Lastde | 91.75 | 84.12 | 87.26 | 81.02 | 71.32 | 60.46 | 18.77 | 0.00 | 61.46 | 55.96 | 66.11 | 56.31 |
| *Gemma-2-2B family (Gemma-2-2B-it+Gemma-2-2B)* | | | | | | | | | | | | |
| Binoculars | 97.19 | 91.76 | 93.27 | 88.30 | 79.39 | 73.27 | 66.54 | 57.00 | 78.59 | 72.55 | 83.00 | 76.58 |
| IRM | 89.13 | 80.01 | 88.12 | 77.83 | 85.50 | 77.67 | 85.44 | 74.45 | 92.70 | 85.13 | 88.38 | 79.02 |
| *Llama-3.2-1B-Instruct* | | | | | | | | | | | | |
| Log-Likelihood | 97.02 | 91.66 | 93.84 | 87.75 | 76.08 | 71.06 | 67.48 | 60.02 | 78.81 | 71.08 | 82.65 | 76.31 |
| Rank | 86.85 | 76.62 | 85.57 | 76.43 | 72.80 | 62.55 | 18.43 | 00.00 | 54.55 | 46.14 | 63.60 | 52.35 |
| Log-Rank | 97.06 | 92.08 | 93.49 | 87.83 | 76.40 | 70.75 | 67.16 | 56.74 | 78.66 | 69.46 | 82.55 | 75.37 |
| LRR | 94.55 | 87.67 | 89.94 | 83.10 | 74.92 | 67.65 | 63.73 | 52.90 | 74.67 | 61.95 | 79.56 | 70.65 |
| Fast-DetectGPT | 63.94 | 56.31 | 60.30 | 53.76 | 69.95 | 63.01 | 72.12 | 65.21 | 81.54 | 74.05 | 69.58 | 62.47 |
| Lastde | 87.30 | 79.51 | 82.70 | 74.39 | 74.29 | 64.53 | 31.80 | 66.69 | 71.86 | 61.99 | 69.59 | 69.42 |
| *Llama-3.2-1B family (Llama-3.2-1B-Instruct+Llama-3.2-1B)* | | | | | | | | | | | | |
| Binoculars | **99.15** | **96.10** | 96.52 | 91.94 | 90.56 | 83.01 | 88.62 | 81.60 | 90.37 | 82.46 | 93.04 | 87.02 |
| IRM | 98.94 | 95.52 | **97.70** | **93.02** | **98.00** | **92.57** | **95.65** | **88.65** | **95.68** | **90.28** | **97.19** | **92.01** |
| *Reward Models* | | | | | | | | | | | | |
| ReMoDetect | 97.12 | 91.00 | 93.47 | 84.59 | 88.59 | 73.86 | 88.74 | 80.44 | 88.19 | 80.55 | 91.22 | 82.09 |
| RM-Deberta-v3-large-v2 | 81.88 | 73.10 | 76.54 | 65.28 | 62.23 | 45.66 | 70.96 | 70.44 | 65.11 | 62.74 | 71.34 | 63.44 |
| GRM-gemma2-2B | 57.93 | 56.02 | 56.33 | 52.33 | 46.54 | 66.68 | 47.42 | 63.66 | 45.20 | 61.15 | 50.69 | 59.97 |
| GRM-Llama3.2-3B | 54.84 | 46.79 | 53.77 | 41.73 | 44.56 | 34.62 | 39.87 | 6.92 | 49.75 | 23.95 | 48.56 | 30.80 |

Table 11: The performance of various detection methods in the human writing task. The best results are marked in **bold** and the second best results are marked by underline.

| Metrics → | AUROC | $F_1$ | AUROC | $F_1$ | AUROC | $F_1$ | AUROC | $F_1$ | AUROC | $F_1$ |
|---|---|---|---|---|---|---|---|---|---|---|
| Attack Settings → | Direct | | Paraph. | | Perturb | | Mixing | | Avg. | |
| *Gemma-2-2B-it* | | | | | | | | | | |
| Log-Likelihood | 94.96 | 88.15 | 83.10 | 76.82 | 99.78 | 98.56 | 93.28 | 86.64 | 92.78 | 87.54 |
| Rank | 79.27 | 69.84 | 70.44 | 64.51 | 97.48 | 93.54 | 69.03 | 59.18 | 79.05 | 71.77 |
| Log-Rank | 95.25 | 88.56 | **83.20** | 78.49 | 99.81 | 98.60 | 93.80 | 86.53 | 93.02 | 88.05 |
| LRR | 93.16 | 85.08 | 81.79 | 78.78 | 99.41 | 96.66 | 91.27 | 82.49 | 91.41 | 85.75 |
| Fast-DetectGPT | 90.59 | 83.82 | 77.95 | 71.34 | 98.61 | 94.92 | 91.09 | 84.37 | 89.56 | 83.61 |
| Lastde | 91.75 | 84.12 | 78.53 | 75.68 | 99.66 | 97.53 | 92.62 | 85.20 | 90.64 | 85.63 |
| *Gemma-2-2B family (Gemma-2-2B-it+Gemma-2-2B)* | | | | | | | | | | |
| Binoculars | 97.19 | 91.77 | 81.44 | 77.73 | 99.85 | 98.51 | 94.67 | 88.28 | 93.29 | 89.07 |
| IRM | 89.13 | 80.02 | 64.60 | 60.92 | 75.28 | 74.29 | 95.77 | 88.92 | 81.19 | 76.04 |
| *Llama-3.2-1B-Instruct* | | | | | | | | | | |
| Log-Likelihood | 97.03 | 91.67 | 83.00 | 78.12 | 99.95 | 99.26 | 96.11 | 90.34 | 94.02 | 89.85 |
| Rank | 86.65 | 76.62 | 70.69 | 70.07 | 99.55 | 97.52 | 80.58 | 68.60 | 84.37 | 78.20 |
| Log-Rank | 97.06 | 92.08 | 82.78 | 78.16 | **99.96** | **99.45** | 95.98 | 89.52 | 93.94 | 89.81 |
| LRR | 94.55 | 87.67 | 80.36 | 77.37 | 99.79 | 97.91 | 91.80 | 83.91 | 91.62 | 86.72 |
| Fast-DetectGPT | 63.95 | 56.32 | 52.58 | 42.72 | 47.42 | 40.47 | 79.36 | 72.98 | 60.83 | 53.12 |
| Lastde | 87.30 | 79.51 | 73.02 | 70.37 | 97.98 | 93.72 | 91.79 | 84.38 | 87.52 | 82.00 |
| *Llama-3.2-1B family (Llama-3.2-1B-Instruct+Llama-3.2-1B)* | | | | | | | | | | |
| Binoculars | **99.16** | **96.10** | 81.25 | **78.81** | 99.84 | 98.56 | 98.26 | 93.82 | **94.63** | **91.82** |
| IRM | 98.94 | 95.52 | 80.84 | 76.22 | 99.35 | 96.33 | **98.79** | **95.06** | 94.48 | 90.78 |
| *Reward Models* | | | | | | | | | | |
| ReMoDetect | 97.12 | 91.00 | 76.02 | 74.91 | 99.51 | 97.09 | 93.89 | 87.59 | 91.63 | 87.65 |
| RM-Deberta-v3-large-v2 | 81.88 | 73.10 | 78.40 | 75.71 | 88.37 | 79.06 | 77.29 | 70.60 | 81.48 | 74.62 |
| GRM-gemma2-2B | 57.94 | 56.03 | 68.50 | 61.71 | 69.80 | 65.10 | 58.53 | 58.63 | 63.69 | 60.37 |
| GRM-Llama3.2-3B | 54.84 | 46.79 | 63.65 | 59.71 | 65.91 | 59.13 | 58.44 | 52.93 | 60.71 | 54.64 |

# B Additional Experiments

## B.1 Additional Benchmarks

**Settings.** We further evaluate our method on the RAID [35] and DivScore [36] benchmarks. We adopt LLaMA-3.2-1B as the backbone detector, which demonstrated strong performance on the DetectRL benchmark. For the RAID benchmark, due to the large scale, we randomly sample 512 annotated pairs from the training split for each source model. We report AUROC as the main evaluation metric. For the DivScore benchmark, we use the official test split for evaluation and report both AUROC and $F_1$ Score. Since the lengths of test samples vary considerably, we compute the mean of implicit rewards across all tokens, instead of the sum (Eq. 5), to normalize the decision threshold, which empirically shows an approximately linear relationship with text length (see Figure 4).

Table 12: Results on RAID benchmark.

| Method | chatgpt | gpt4 | gpt3 | gpt2 | llama-chat | mistral | mistral-chat | mpt | mpt-chat | cohere | cohere-chat |
|---|---|---|---|---|---|---|---|---|---|---|---|
| Log-Likelihood | 98.48 | 93.51 | 97.35 | 68.77 | 98.93 | 69.82 | 97.78 | 48.84 | 80.08 | 89.26 | 92.42 |
| Log-Rank | 98.71 | 92.95 | 96.94 | 71.13 | 99.12 | 71.34 | 98.03 | 51.83 | 81.27 | 87.43 | 91.89 |
| Binoculars | 99.66 | 96.38 | 99.12 | 79.30 | 99.05 | 72.45 | 98.22 | 51.99 | 83.41 | 97.06 | 98.00 |
| IRM | 99.58 | 99.27 | 87.43 | 86.31 | 99.99 | 62.25 | 99.35 | 57.57 | 92.42 | 87.14 | 92.10 |
| ReMoDetect | 99.69 | 97.82 | 94.57 | 57.36 | 99.33 | 59.18 | 97.70 | 43.06 | 89.63 | 85.29 | 93.02 |

Table 13: Results on DivScore benchmark.

| Method | DeepSeek-R1 | | DeepSeek-V3 | | GPT-4o | | O3-mini | | Avg. | |
|---|---|---|---|---|---|---|---|---|---|---|
| | AUROC | F1 | AUROC | F1 | AUROC | F1 | AUROC | F1 | AUROC | F1 |
| Log-Likelihood | 88.64 | 86.11 | 98.87 | 97.68 | 85.96 | 81.50 | 90.37 | 87.47 | 90.96 | 88.19 |
| Log-Rank | 87.56 | 84.85 | 98.91 | 97.69 | 86.28 | 82.34 | 90.86 | 88.01 | 90.90 | 88.22 |
| Binoculars | 98.03 | 94.18 | 99.87 | 99.03 | 94.84 | 90.74 | 95.61 | 91.88 | 97.09 | 93.96 |
| IRM | 97.39 | 93.67 | 99.27 | 97.77 | 99.08 | 96.74 | 97.95 | 94.25 | 98.42 | 96.61 |
| ReMoDtect | 95.86 | 90.69 | 99.73 | 98.20 | 99.56 | 97.46 | 98.73 | 94.73 | 98.47 | 95.27 |

**Results.** As shown in Table 12 and Table 13, IRM performs competitively across a wide range of models, supporting the generalizability of our approach.

## B.2 Additional Baselines

**Settings.** We further include Skywork-Reward-V2 [37] models as additional baselines. These reward models are trained on large-scale preference data and use backbones up to 8B parameters. They achieve SOTA on the RewardBench benchmark, providing a strong reference to evaluate whether reward model scaling enhances generalization in LLM-generated text detection.

Table 14: Comparison of IRM and Skywork-Reward-V2 on the DetectRL benchmark.

| Tasks Settings → | Multi-Domain | | Multi-LLM | | Multi-Attack | | Generalization | | | Length | | Human Writing | | Avg. |
|---|---|---|---|---|---|---|---|---|---|---|---|---|---|---|
| Methods ↓ | AUROC | $F_1$ | AUROC | $F_1$ | AUROC | $F_1$ | Domain $F_1$ | LLM $F_1$ | Attack $F_1$ | Train $F_1$ | Test $F_1$ | AUROC | $F_1$ | |
| *Llama-3.1-8B* | | | | | | | | | | | | | | |
| Skywork-Reward-V2 | 71.33 | 62.97 | 70.67 | 66.15 | 68.23 | 65.85 | 59.06 | 65.59 | 63.23 | 45.41 | 61.78 | 39.32 | 23.98 | 58.74 |
| IRM | 90.52 | 84.67 | 89.01 | 83.18 | 88.70 | 82.13 | 81.10 | 79.36 | 81.47 | 74.28 | 76.78 | 82.61 | 76.18 | 82.31 |
| *Llama-3.2-3B* | | | | | | | | | | | | | | |
| Skywork-Reward-V2 | 63.48 | 48.47 | 63.33 | 57.78 | 60.32 | 41.35 | 44.50 | 56.99 | 32.58 | 34.77 | 49.65 | 30.52 | 6.91 | 45.43 |
| IRM | 97.31 | 92.51 | 97.25 | 92.15 | 97.21 | 91.69 | 91.68 | 90.67 | 91.37 | 78.59 | 80.97 | 93.85 | 88.85 | 91.08 |
| *Qwen3-8B* | | | | | | | | | | | | | | |
| Skywork-Reward-V2 | 68.21 | 57.01 | 67.94 | 62.23 | 66.08 | 55.31 | 54.21 | 61.97 | 54.88 | 40.19 | 44.90 | 35.90 | 19.82 | 52.97 |
| IRM | 74.42 | 58.02 | 73.08 | 59.05 | 73.70 | 62.13 | 55.77 | 57.70 | 61.15 | 42.54 | 41.77 | 69.61 | 58.49 | 60.57 |

**Results.** As shown in Table 14, IRM consistently outperforms the reward-model-based methods, even when using the same or smaller backbones. This further supports our claim that standard preference-trained reward models, despite their size and strong performance on preference classification, do not generalize well to the LLM-generated text detection task. In contrast, IRM's implicit reward modeling is inherently aligned with this detection objective, leading to robust performance across different settings.

---
**Algorithm 1** Inference pipeline of IRM
---
**Inputs:** policy model $\pi_\theta$, reference model $\pi_{\text{ref}}$, detected text $x$, decision threshold $t$.
**Output:** authenticity label $y \in \{\text{human-written}, \text{LLM-generated}\}$

1: $p_{\text{ref}} \leftarrow \pi_{\text{ref}}(x)$
2: $p_\theta \leftarrow \pi_\theta(x)$
3: $r \leftarrow \log p_\theta - \log p_{\text{ref}}$                 ▷ Compute implicit reward score
4: **if** $r < t$ **then**
5:      $y \leftarrow$ human-written
6: **else**
7:      $y \leftarrow$ LLM-generated
8: **end if**
9: **return** $y$
---

