# OpenReview forum: "Zero-Shot Detection of LLM-Generated Text via Implicit Reward Model"
_NeurIPS.cc/2025/Conference — NeurIPS 2025 poster_

### Official Review · Reviewer_FXJi · 2025-07-02

**Clarity:** 3
**Significance:** 3
**Originality:** 3
**Rating:** 5
**Confidence:** 4

**Summary:**

This paper proposes a novel, lightweight approach to detect AI-generated texts. The proposed method leverages the implicit reward model constructed from the relation between internal representations of base and instruction-tuned versions of an LLM theoretically backed by the properties of RLHF finetuning. Using it, authors develop a detector of artificial texts. Conducted experiments show its high performance against various generating models and adversarial attacks on different text domains (styles/genres).

**Questions:**

See Weaknesses.

**Ethical Concerns:**

["NO or VERY MINOR ethics concerns only"]

**Final Justification:**

The reply to my review answers my concerns regarding the proposed method and its evaluation. I have also read responses to other reviews.

In their replies, authors provide numerous new results showing high performance of the proposed method against various modern generating models; they include evaluation on more benchmarks (e.g., RAID, DivScore), and additional quality metrics were provided.

Considering the incorporation of the new results and proposed changes, I think that this paper can be accepted (and I raise my overall score for it from 4 to 5).

**Limitations:**

Yes

**Paper Formatting Concerns:**

- Axis/tick labels on Figures 1, 3 are a little difficult to read.

**Quality:**

2

**Strengths And Weaknesses:**

Strengths:

- An interesting approach leveraging the RLHF-nature of the common model instruction fine-tuning scheme. Proposed method achieves noticeable improvements over competitive solutions and is relatively easy to deploy.

- Experiments are quite extensive, showing the robustness of the proposed AI-text detector to domain shift and various adversarial attacks. Evaluation also covers some less common attacks (e.g., translation of a text generated in another language).

- The paper is well-structured and processing in a logically coherent manner.


Weaknesses:

- My main concerns with this paper are particular limitations of the evaluation of the proposed method. First of all, only one dataset is used - DetectRL - and it covers models primarily from the first half/mid-2023. Some more modern models (e.g., GPT4-o or something else) would be appreciated.
Secondly, evaluation metrics. In practice, type I and type II errors (false positive and false negative) have different weights. None of the metrics provided for the experiments (AUROC and $F_1$) cover this aspect; I would suggest providing FPR/FNR (or any other appropriate metric).
Also, I would like to ask if it is possible to include metrics for baselines in Table 3?

- On another note, a clearly outlined pipeline for application of the proposed method to a text (possibly, an example) in the paper (appendices) would be much appreciated.

---

> ### Author Rebuttal · Authors · 2025-07-30
>
> > Weaknesses1: My main concerns with this paper are particular limitations of the evaluation of the proposed method. First of all, only one dataset is used - DetectRL - and it covers models primarily from the first half/mid-2023. Some more modern models (e.g., GPT4-o or something else) would be appreciated.
>
> Thank you for raising this important point. We agree that evaluating detectors on outputs from more modern models is crucial, especially given their increasing prevalence in real-world use cases.
>
> To address this, we conducted additional experiments on the DivScore [1] benchmark, which includes four professionally legal and medical datasets generated by more modern LLMs: DeepSeek-R1, DeepSeek-V3, GPT-4o, and O3-mini. For evaluation, we use the main test split of DivScore and adopt LLaMA-3.2-1B as the backbone, which demonstrated strong performance on DetectRL. In addition to AUROC and F1, we also report false positive rate (FPR) and false negative rate (FNR), as these provide a more nuanced understanding of detection quality in high-stakes settings.
>
> | Method         | DeepSeek-R1 |       | DeepSeek-V3 |       | GPT-4o |       | O3-mini |       | Avg.  |       |
> |----------------|-------------|-------|-------------|-------|--------|-------|---------|-------|-------|-------|
> |                | AUROC       | F1    | AUROC       | F1    | AUROC  | F1    | AUROC   | F1    | AUROC | F1    |
> | Log-Likelihood | 88.64       | 86.11 | 98.87       | 97.68 | 85.96  | 81.50 | 90.37   | 87.47 | 90.96 | 88.19 |
> | Log-Rank       | 87.56       | 84.85 | 98.91       | 97.69 | 86.28  | 82.34 | 90.86   | 88.01 | 90.90 | 88.22 |
> | Binoculars     | 98.03       | 94.18 | 99.87       | 99.03 | 94.84  | 90.74 | 95.61   | 91.88 | 97.09 | 93.96 |
> | IRM            | 97.39       | 93.67 | 99.27       | 97.77 | 99.08  | 96.74 | 97.95   | 94.25 | 98.42 | 96.61 |
> | ReMoDtect      | 95.86       | 90.69 | 99.73       | 98.20 | 99.56  | 97.46 | 98.73   | 94.73 | 98.47 | 95.27 |
>
> | Method         | DeepSeek-R1 |       | DeepSeek-V3 |      | GPT-4o |       | O3-mini |      | Avg.  |      |
> |----------------|-------------|-------|-------------|------|--------|-------|---------|------|-------|------|
> |                | FPR         | FNR   | FPR         | FNR  | FPR    | FNR   | FPR     | FNR  | FPR   | FNR  |
> | Log-Likelihood | 21.44       | 9.19  | 3.23        | 1.50 | 15.00  | 19.02 | 17.48   | 9.11 | 14.29 | 9.71 |
> | Log-Rank       | 21.95       | 11.01 | 3.04        | 1.66 | 15.47  | 17.57 | 16.22   | 9.18 | 14.17 | 9.86 |
> | Binoculars     | 6.42        | 5.31  | 1.00        | 0.93 | 5.60   | 11.82 | 10.33   | 6.37 | 5.84  | 6.11 |
> | IRM            | 7.26        | 5.53  | 1.57        | 2.82 | 2.72   | 3.75  | 5.05    | 6.34 | 4.15  | 4.61 |
> | ReMoDtect      | 11.46       | 7.58  | 2.16        | 1.45 | 2.37   | 2.67  | 6.39    | 4.25 | 5.60  | 3.99 |
>
> Across both tables, IRM demonstrates consistently strong performance on recent reasoning models, achieving high AUROC and F1 while also maintaining low FPR and FNR. These results support the robustness and reliability of IRM for detecting high-quality responses from modern reasoning LLMs.
>
> > Weaknesses2: Secondly, evaluation metrics. In practice, type I and type II errors (false positive and false negative) have different weights. None of the metrics provided for the experiments (AUROC and $F_1$) cover this aspect; I would suggest providing FPR/FNR (or any other appropriate metric).
>
> Thank you for this valuable suggestion. While the standard evaluation protocol of the DetectRL benchmark does not include FPR/FNR metrics, we agree that false positive and false negative rates are important in practical applications. To address this, we report below the FPR and FNR for our method and competitive baselines on the multi-LLM sub-task of the DetectRL benchmark, using Llama-3.2-1B as the backbone.
>
> | Method         | GPT-3.5 |       | Claude |       | PaLM-2 |       | Llama-2 |       | Avg.  |       |
> |----------------|---------|-------|--------|-------|--------|-------|---------|-------|-------|-------|
> |                | FPR     | FNR   | FPR    | FNR   | FPR    | FNR   | FPR     | FNR   | FPR   | FNR   |
> | Log-Likelihood | 11.60   | 29.86 | 7.60   | 67.96 | 8.20   | 33.43 | 6.50    | 19.64 | 8.47  | 37.72 |
> | Log-Rank       | 6.50    | 36.41 | 10.40  | 64.68 | 7.30   | 33.83 | 7.50    | 18.25 | 7.92  | 38.29 |
> | Binoculars     | 7.80    | 20.54 | 19.90  | 29.07 | 4.80   | 12.70 | 2.90    | 4.07  | 8.85  | 16.59 |
> | IRM            | 5.30    | 9.92  | 11.00  | 9.62  | 7.50   | 12.30 | 2.20    | 4.56  | 6.50  | 9.10  |
> | ReMoDetect     | 7.30    | 13.59 | 44.70  | 10.12 | 27.50  | 16.87 | 8.20    | 15.58 | 21.92 | 14.04 |
>
> As shown, IRM consistently achieves lower FPR and FNR across models.
>
> > Weaknesses3: Also, I would like to ask if it is possible to include metrics for baselines in Table 3?
>
> Thank you for the suggestion. We provide below an extended Table 3 with detailed results for competitive baselines.
>
> | Method     | DIPPER Paraphrase |       |       |       | Polish using LLMs |       |       |       | Back Translation |       |       |       |
> |------------|-------------------|-------|-------|-------|-------------------|-------|-------|-------|------------------|-------|-------|-------|
> |            | Pre               | Rec   | $F_1$ | AUROC | Pre               | Rec   | $F_1$ | AUROC | Pre              | Rec   | $F_1$ | AUROC |
> | Log-Likelihood | 92.57             | 84.03 | 88.09 | 93.76 | 85.79             | 76.69 | 80.98 | 88.43 | 99.01            | 99.01 | 99.01 | 99.87 |
> | Log-Rank    | 92.52             | 84.62 | 88.39 | 93.92 | 87.03             | 75.89 | 81.08 | 88.66 | 99.01            | 99.31 | 99.16 | 99.90 |
> | Binoculars | 88.67             | 81.55 | 84.96 | 92.14 | 82.89             | 81.25 | 82.06 | 89.39 | 98.59            | 96.83 | 97.70 | 99.70 |
> | IRM        | 93.02             | 83.33 | 87.91 | 94.39 | 77.76             | 79.76 | 78.75 | 85.36 | 93.11            | 87.10 | 90.01 | 95.54 |
> | ReMoDetect | 95.30             | 86.41 | 90.63 | 96.35 | 80.00             | 56.35 | 66.12 | 77.50 | 99.40            | 98.31 | 98.85 | 99.81 |
>
> > Weaknesses4: On another note, a clearly outlined pipeline for application of the proposed method to a text (possibly, an example) in the paper (appendices) would be much appreciated.
>
> Thank you for the suggestion. We agree that a clearly outlined pipeline would improve the clarity and usability of our method. In the final version, we will include a figure that illustrates the full application process of IRM with a step-by-step example. This figure will demonstrate how to compute the implicit reward scores from a given text and how these scores are used to make the final classification decision.
>
> [1] DivScore: Zero-Shot Detection of LLM-Generated Text in Specialized Domains

---

> > ### Comment · Reviewer_FXJi · 2025-08-06
> >
> > Thank you for your extensive reply; it answers my concerns regarding the proposed method and its evaluation. I have also read responses to other reviews. Considering new results and proposed changes, I will raise my score for this paper.

---

### Official Review · Reviewer_zjGC · 2025-07-02

**Clarity:** 4
**Significance:** 3
**Originality:** 3
**Rating:** 5
**Confidence:** 4

**Summary:**

This paper introduces a new framework Implicit Reward Model (IRM), a training-free LLM-generated text (LGT) detector which is motivated by prior work ReMoDetect. The method utilized implicit reward implied by Direct Preference Optimization (DPO). Experiments conducted on detection benchmark (DetectRL) show that IRM significantly outperforms other baselines. Also, the paper shows the robustness of IRM against the domain, LLM, attack, and text-length shifts.

**Questions:**

[Q1] Detecting recent reasoning models: Have you evaluated IRM and other baselines with outputs from reasoning models such as OpenAI o3 or Deepseek-R1?

[Q2] Can you provide evaluation results on detecting randomly sampled generations by adjusting temperatures?

**Ethical Concerns:**

["NO or VERY MINOR ethics concerns only"]

**Final Justification:**

After reading the authors' rebuttal and further discussions with the authors, I have decided to maintain my score of 5. My main justification is that most of my initial concerns have been addressed, although some issues remain.

**Resolved**

1. Evaluation on recent Large Reaasoning Models (LRMs): The authors introduced a new detection evaluation and showed that IRM outperforms the baselines. They also provided interesting observations that some metric-based detection methods are not robust for LRMs, whereas IRM remains robust.

**Unresolved**

1. Robustness of Detection under Varying Generation Temperatures: While the authors noted the lack of significance in this evaluation and cited time constraints, this concern remains unresolved.

**Limitations:**

yes

**Paper Formatting Concerns:**

No issues were founded.

**Quality:**

4

**Strengths And Weaknesses:**

[S1] Great detection performance and generalization: IRM consistently outperform baselines.

[S2] No further training needed: The detector don’t need further training which significantly

[S3] Extensive experiments: Contains extensive evaluation across various benchmarks and baselines.

[S4] Insightful Analysis: The paper quantifies threshold drift with length, explores model version and size effects.

[W1] Didn't analyzed about detecting reasoning models: DetectRL doesn’t contains recent reasoning models’ responses such as openai o3 or deepseek-r1.

---

> ### Author Rebuttal · Authors · 2025-07-30
>
> > Weaknesses 1 and Questions1: Didn't analyzed about detecting reasoning models: DetectRL doesn’t contains recent reasoning models’ responses such as openai o3 or deepseek-r1. Detecting recent reasoning models: Have you evaluated IRM and other baselines with outputs from reasoning models such as OpenAI o3 or Deepseek-R1?
>
> Thank you for raising this important point. We agree that evaluating detectors on outputs from strong reasoning models is crucial, especially given their increasing prevalence in real-world use cases.
>
> To address this, we conducted additional experiments on the DivScore [1] benchmark, which includes four professionally legal and medical datasets generated by recent reasoning-oriented LLMs: DeepSeek-R1, DeepSeek-V3, GPT-4o, and O3-mini. For evaluation, we use the main test split of DivScore and adopt LLaMA-3.2-1B as the backbone, which demonstrated strong performance on DetectRL. In addition to AUROC and F1, we also report false positive rate (FPR) and false negative rate (FNR), as these provide a more nuanced understanding of detection quality in high-stakes settings.
>
> | Method         | DeepSeek-R1 |       | DeepSeek-V3 |       | GPT-4o |       | O3-mini |       | Avg.  |       |
> |----------------|-------------|-------|-------------|-------|--------|-------|---------|-------|-------|-------|
> |                | AUROC       | F1    | AUROC       | F1    | AUROC  | F1    | AUROC   | F1    | AUROC | F1    |
> | Log-Likelihood | 88.64       | 86.11 | 98.87       | 97.68 | 85.96  | 81.50 | 90.37   | 87.47 | 90.96 | 88.19 |
> | Log-Rank       | 87.56       | 84.85 | 98.91       | 97.69 | 86.28  | 82.34 | 90.86   | 88.01 | 90.90 | 88.22 |
> | Binoculars     | 98.03       | 94.18 | 99.87       | 99.03 | 94.84  | 90.74 | 95.61   | 91.88 | 97.09 | 93.96 |
> | IRM            | 97.39       | 93.67 | 99.27       | 97.77 | 99.08  | 96.74 | 97.95   | 94.25 | 98.42 | 96.61 |
> | ReMoDtect      | 95.86       | 90.69 | 99.73       | 98.20 | 99.56  | 97.46 | 98.73   | 94.73 | 98.47 | 95.27 |
>
> | Method         | DeepSeek-R1 |       | DeepSeek-V3 |      | GPT-4o |       | O3-mini |      | Avg.  |      |
> |----------------|-------------|-------|-------------|------|--------|-------|---------|------|-------|------|
> |                | FPR         | FNR   | FPR         | FNR  | FPR    | FNR   | FPR     | FNR  | FPR   | FNR  |
> | Log-Likelihood | 21.44       | 9.19  | 3.23        | 1.50 | 15.00  | 19.02 | 17.48   | 9.11 | 14.29 | 9.71 |
> | Log-Rank       | 21.95       | 11.01 | 3.04        | 1.66 | 15.47  | 17.57 | 16.22   | 9.18 | 14.17 | 9.86 |
> | Binoculars     | 6.42        | 5.31  | 1.00        | 0.93 | 5.60   | 11.82 | 10.33   | 6.37 | 5.84  | 6.11 |
> | IRM            | 7.26        | 5.53  | 1.57        | 2.82 | 2.72   | 3.75  | 5.05    | 6.34 | 4.15  | 4.61 |
> | ReMoDtect      | 11.46       | 7.58  | 2.16        | 1.45 | 2.37   | 2.67  | 6.39    | 4.25 | 5.60  | 3.99 |
>
>
>
> Across both tables, IRM demonstrates consistently strong performance on recent reasoning models, achieving high AUROC and F1 while also maintaining low FPR and FNR. These results support the robustness and reliability of IRM for detecting high-quality responses from modern reasoning LLMs.
>
>
>
> > Questions2: Can you provide evaluation results on detecting randomly sampled generations by adjusting temperatures?
>
> Thank you for this insightful question. Sampling temperature indeed has a significant influence on the quality and diversity of LLM-generated outputs. However, existing benchmarks, such as DetectRL, typically fix the temperature at a scalar value (e.g., 1.0) and do not explore how detection performance varies across temperature settings.
>
> This design choice is grounded in practical usage. In many real-world applications, such as ChatGPT or similar LLM-based systems, users cannot directly control temperature or generation strategy, and the outputs are produced under a fixed decoding setup predefined by the provider. Therefore, the kinds of generations actually presented to users are restricted to a narrow range of temperature settings.
>
> [1] DivScore: Zero-Shot Detection of LLM-Generated Text in Specialized Domains

---

> > ### Comment · Reviewer_zjGC · 2025-08-04
> >
> > Thank you for your rebuttal. Most of my concerns regarding the robustness of the method have been addressed. However, I have a few follow-up questions.
> >
> > ---
> >
> > **[W1,Q1]**
> > Thank you for the detailed experiments comparing various recent Large Language Models (LLMs) and Large Reasoning Models (LRMs). I agree that the IRM shows impressive performance even without training.
> >
> > However, I am curious about the observation that while zero-shot metric-based methods (e.g., Log-Likelihood, Log-Rank) perform well on LLMs, they seem to perform poorly on LRMs. My initial thought is that this could be due to a distribution shift between the questions and answers, especially since most reasoning models do not expose their reasoning steps to end users.
> >
> > Could the authors please discuss or provide some intuition as to why this discrepancy occurs?
> >
> > ---
> >
> > **[Q2]**
> > I agree that most chat-based services do not allow flexible control over the temperature. However, some API-based services [1,2] do provide the ability to vary temperature. I believe some of use-cases are using API-based services, so the robustness over temperature could be important.
> >
> > I understand there may not have been enough time to construct a new dataset beyond the existing benchmark, but I would appreciate it if you could consider including such experiments in the final version.
> >
> > ---
> >
> > [1] https://platform.openai.com/docs/api-reference/chat/create \
> > [2] https://docs.anthropic.com/en/api/messages

---

> > > ### Author Response · Authors · 2025-08-04
> > >
> > > > [W1,Q1] Performance Discrepancy of Zero-shot Metric-based Methods (LLMs vs LRMs)
> > >
> > > Thank you for your insightful question. We agree with your hypothesis that the discrepancy may stem from a distribution shift between questions and answers. To further investigate this, we conduct an additional experiment on the DivScore benchmark, which includes llm-reasoning texts from DeepSeek-R1. This allows us to directly test whether incorporating reasoning content improves detection performance. Specifically, we prepend the reasoning content to the inputs and then re-evaluate detection methods.
> > >
> > > | Method         | without reasoning |       |       |       | with reasoning |       |       |      |
> > > |----------------|-------------------|-------|-------|-------|----------------|-------|-------|------|
> > > |                | AUROC             | F1    | FPR   | FNR   | AUROC          | F1    | FPR   | FNR  |
> > > | Log-Likelihood | 88.64             | 86.11 | 21.44 | 9.19  | 92.31          | 90.59 | 15.45 | 5.14 |
> > > | Log-Rank       | 87.56             | 84.85 | 21.95 | 11.01 | 92.91          | 90.90 | 15.13 | 4.77 |
> > > | Binoculars     | 98.03             | 94.18 | 6.42  | 5.31  | 98.95          | 97.08 | 4.71  | 1.27 |
> > > | IRM            | 97.39             | 93.67 | 7.26  | 5.53  | 99.77          | 98.88 | 1.40  | 0.85 |
> > > | ReMoDtect      | 95.86             | 90.69 | 11.46 | 7.58  | 90.88          | 85.99 | 19.72 | 9.81 |
> > >
> > > As shown in above Table, including reasoning content significantly improves the performance of Log-Likelihood and Log-Rank, supporting your intuition. Interestingly, IRM and Binoculars also benefit from the added reasoning content, achieving even better performance. In contrast, ReMoDtect shows performance degradation, possibly due to the distribution mismatch between the reasoning format and its training set.
> > >
> > > -------
> > >
> > > > [Q2] Robustness of Detection under Varying Generation Temperatures
> > >
> > > Thank you for your thoughtful suggestion. We agree that temperature can influence model generations, and we will seriously consider incorporating such experiments in the final version of this work.
> > >
> > > That said, we would like to clarify a few practical challenges. The API links you kindly provided ([1], [2]) correspond to OpenAI and Anthropic services. However, the reasoning-capable models provided by these platforms (e.g., o3-mini, Claude 3.7 and 4) currently do not support temperature control for end users. For instance, [3] explicitly states that “Thinking isn’t compatible with temperature or top\_k modifications as well as forced tool use.” Similarly, in the openai-python library, a relevant issue regarding temperature control for o3-mini has been marked as “Closed as not planned” [4].
> > >
> > > Furthermore, generation temperature can have model-specific and task-specific effects. That is, each model may require a different temperature range to yield high-quality outputs. As a result, it is not meaningful to directly compare or aggregate detection performance across models at fixed temperatures, since what constitutes a "reasonable" generation may vary significantly by model. This introduces not only substantial computational cost, due to the need for exhaustive grid searches, but also a fundamental challenge in experimental design, as there is no standardized normalization or alignment across models under varying temperatures.
> > >
> > > Nonetheless, we appreciate the direction suggested and will explore practical ways to systematically evaluate robustness to generation temperature in future work.
> > >
> > > -------
> > >
> > > [3] Building with extended thinking
> > >
> > > [4] temperature is not supported with this model(o3-mini) \#2104

---

> > > > ### Comment · Reviewer_zjGC · 2025-08-04
> > > >
> > > > Thank you for your detailed discussion. I have no further questions. I believe this paper will give a significant contribution to the NeurIPS community by offering interesting perspectives and robust evaluations. For this reason, I will maintain my score as "accept" (5).

---

### Official Review · Reviewer_NPcg · 2025-07-03

**Clarity:** 2
**Significance:** 2
**Originality:** 2
**Rating:** 4
**Confidence:** 3

**Summary:**

This paper introduces IRM, a zero-shot text detector. It scores a text via the DPO reward model parameterization, i.e. the log-ratio between an instruction-tuned model and its corresponding base model.

**Questions:**

- On Table 5, as the paper states, "larger or newer versions of a model do not always lead to better detection performance." Do you have any interpretation of why this is the case? Intuitively, it seems that the implicit reward model from a larger backbone should be better at estimating reward, which should, in turn, improve its ability to recognize "overoptimized text."
- Have you tried larger models, such as 7b models or beyond? I'd be interested to see whether the scaling trends continue to hold.
- Could you try the same method with other types of reward models based on the same backbones (without further fine-tuning)? For example, https://huggingface.co/Ray2333/GRM-gemma2-2B-rewardmodel-ft this model uses the gemma2-2b backbone.

**Ethical Concerns:**

["NO or VERY MINOR ethics concerns only"]

**Final Justification:**

I am raising my score. The authors addressed my concerns, particularly the one around motivation for focusing on implicit RMs.

**Limitations:**

- Tests on quite small models.
- Motivation for considering implicit reward models is unclear.

**Quality:**

3

**Strengths And Weaknesses:**

- The method has a simple implementation and yields significant gains. It requires no extra data, fine-tuning, or text perturbation. The motivation is clear: machine-generated text typically overoptimizes reward during RLHF, so reward can be a criterion for distinguishing btw human and machine text.
- The paper provides thorough ablations on text length and robustness.
- The motivation for specifically focusing on implicit reward models is unclear to me. Aside from DPO-style reward models, you could use any available reward model for this purpose (sequence classifier or even generative reward models). There are a bunch in [1]. Do you have a reason for focusing on implicit reward models specifically?

[1] https://huggingface.co/spaces/allenai/reward-bench

---

> ### Author Rebuttal · Authors · 2025-07-30
>
> > Weaknesses1: The motivation for specifically focusing on implicit reward models is unclear to me. Aside from DPO-style reward models, you could use any available reward model for this purpose (sequence classifier or even generative reward models). There are a bunch in RewardBench. Do you have a reason for focusing on implicit reward models specifically?
>
> Thank you for the insightful question. While it is possible to use various existing reward models such as sequence classifiers or generative reward models, our focus on implicit reward models is motivated by both practical considerations and theoretical alignment with the nature of LLM-generated text.
>
> Specifically, many publicly available reward models are trained for preference classification, not detection of LLM-generated content. These are fundamentally different tasks. Furthermore, reward models used during real-world RLHF pipelines are not publicly released, limiting their practical applicability in detection.
>
> In contrast, DPO-style implicit reward models offer a principled proxy for the unknown reward functions used during RLHF. The log-likelihood difference between an instruction-tuned model and its base model can be interpreted as an implicit reward signal that reflects how much a given response is encouraged during alignment training. This aligns well with the over-optimization nature of LLM-generated text, making it a natural fit for detection tasks.
>
> Finally, we conducted empirical evaluations of several SOTA public reward models trained on preference data (see the rebuttal for Question 3), and observed that they perform poorly on LLM-generated text detection tasks—further validating the effectiveness of our implicit reward model approach.
>
> > Questions1: On Table 5, as the paper states, "larger or newer versions of a model do not always lead to better detection performance." Do you have any interpretation of why this is the case? Intuitively, it seems that the implicit reward model from a larger backbone should be better at estimating reward, which should, in turn, improve its ability to recognize "overoptimized text."
>
> Thank you for raising this insightful question. While larger models tend to be better at estimating reward signals within the distribution they are trained on, the zero-shot detection task inherently involves a cross-distribution generalization challenge. Specifically, IRM is designed to detect whether a given text has been overoptimized with respect to another model’s reward signal—not the preferences of the IRM model itself.
>
>
> In practice, different LLMs are trained on diverse datasets and optimized via different RLHF pipelines. Even when they target similar alignment goals (e.g., helpfulness, factuality), the actual learned preferences can differ significantly due to differences in training data and optimization processes. This phenomenon, which we refer to as reward divergence, means that larger models do not necessarily provide better detection signals—sometimes, a smaller model generalizes better across such divergent reward landscapes.
>
> > Questions2: Have you tried larger models, such as 7b models or beyond? I'd be interested to see whether the scaling trends continue to hold.
>
> Thank you for raising this insightful point. We extend our analysis in Table 5 by evaluating IRM with a range of backbone sizes. The table below reports the results on the DetectRL benchmark:
>
> | Backbone     | Avg.   |
> |--------------|-------|
> | Llama-3.2-1B | 91.77 |
> | Llama-3.2-3B | 91.08 |
> | Llama-3.1-8B | 82.31 |
> | Gemma-2-2B   | 77.05 |
> | Gemma-2-9B   | 64.34 |
>
> These results show that larger backbone models do not necessarily lead to better detection performance in the IRM framework. This trend is particularly evident within model families, where smaller models often outperform their larger counterparts. This supports our claim that IRM's success relies more on cross-model generalization than on reward estimation fidelity, and that larger models may overfit to their own reward landscape, reducing their effectiveness in detecting overoptimization from other models.
>
> > Questions3: Could you try the same method with other types of reward models based on the same backbones (without further fine-tuning)?
>
> Thank you for the suggestion. We conducted additional experiments using public reward models trained on large-scale preference data, specifically the Skywork-Reward-V2 [1] models, which achieve SOTA results on the RewardBench benchmark. These reward models are trained on 40 million preference pairs and use backbones up to 8B in size, allowing us to assess whether reward model scaling improves generalization in LLM-generated text detection.
>
> We compare IRM and Skywork-Reward-V2 under the same backbones, without additional fine-tuning, and report their performance on the DetectRL benchmark.
>
> | Backbone     | Skywork-Reward-V2 | IRM   |
> |--------------|-------------------|-------|
> | Llama-3.1-8B | 58.74             | 82.31 |
> | Llama-3.2-3B | 45.43             | 91.08 |
> | Qwen3-8B     | 52.97             | 60.57 |
>
> These results demonstrate that IRM consistently outperforms the reward-model-based methods, even when using the same or smaller backbones. This further supports our claim that standard preference-trained reward models, despite their size and strong performance on preference classification, do not generalize well to the LLM-generated text detection task. In contrast, IRM’s implicit reward modeling remains robust across settings.
>
> [1] Skywork-Reward-V2: Scaling Preference Data Curation via Human-AI Synergy

---

### Official Review · Reviewer_xWbt · 2025-07-03

**Clarity:** 2
**Significance:** 2
**Originality:** 3
**Rating:** 4
**Confidence:** 3

**Summary:**

This paper proposes IRM, a zero-shot LLM text detector based on implicit reward models. It uses instruction-tuned and base model pairs to compute their log likelihood ratio, which serves as the detector signal. These implicit reward models have the benefit of not needing to train an explicit reward model for the task of LLM detection, and instead directly use pre-trained instruction-tuned and base model pairs, which are generally readily available. Experiments show improved performance compared to other zero-shot LLM detection methods, and directly supervised methods, like the ones that train a direct reward model.

**Questions:**

See weaknesses.

**Ethical Concerns:**

["NO or VERY MINOR ethics concerns only"]

**Final Justification:**

The authors have addressed all of my concerns, so I am raising my score.

**Limitations:**

yes

**Quality:**

3

**Strengths And Weaknesses:**

Strengths
- the paper is well-written and easy to follow
- the method is novel for the text detection scenario. I do not believe any other method has used the log likelihood ratio used in RLHF as a signal for LLM text detection
- Experiments are fairly extensive, covering multiple domains, models, attacks on detectors (such as rewriting).
- A study on the impact of text length is also provided, which has not been extensively reported in prior works, and is helpful to know the limitations of the method

Weaknesses
- Intuitively, if text comes from the preference dataset, shouldn't they have high implicit reward, even if written by a human, since the preference dataset should align more with what humans would write? This seems to contradict the intuition provided in the paper (Figure 1, right side). I am curious how this figure would look if we compared the implicit reward distribution between preference data, non-preference data, for both human and LLM versions. While any detector signal (such as implicit reward) can in theory be used, it would be great to clarify the intuition behind the implicit reward.
- An additional benchmark dataset would help strengthen the paper, such as RAID (https://arxiv.org/abs/2405.07940). While DetectRL covers many scenarios, it would help support the papers claims if at least one other standard benchmark dataset suite is included.
- It would be helpful to understand the effect of LLM size on the detector performance, similar to https://arxiv.org/pdf/2305.09859 for DetectGPT. This would help users understand which instruction-tuned/base model policy pairs and their sizes are the most "universal" across LLMs and datasets..
- Is there a fully-supervised classifier baseline, such as RoBERTa? The fully supervised models seem trained for reward modeling, but I wonder how a baseline that directly tries to classify human/LLM text would compare.

---

> ### Author Rebuttal · Authors · 2025-07-30
>
> > Weakness1: Intuitively, if text comes from the preference dataset, shouldn't they have high implicit reward, even if written by a human, since the preference dataset should align more with what humans would write? This seems to contradict the intuition provided in the paper (Figure 1, right side). I am curious how this figure would look if we compared the implicit reward distribution between preference data, non-preference data, for both human and LLM versions. While any detector signal (such as implicit reward) can in theory be used, it would be great to clarify the intuition behind the implicit reward.
>
> Thank you for the thoughtful comment. We understand the intuition that preferred texts should receive high reward scores, possibly even human-written ones. However, we would like to clarify a key point: in RLHF pipelines, preference datasets consist of multiple LLM-generated responses, which are then ranked by human annotators. They do not include human-written completions labeled as “preferred.” This distinction is critical: the reward model is trained to distinguish between better and worse LLM outputs, not between LLM-generated and human-written texts.
>
> We believe this confusion may stem from conflating “human-preferred” with “human-written.” While human-written texts may be of high quality, they are not optimized to align with the reward model. In contrast, LLM outputs in RLHF are directly optimized to maximize reward scores. This explains the distribution shown in Figure 1 (right): LLM outputs tend to have higher IRM scores than human-written ones, reflecting their alignment with the learned reward model.
>
> This misunderstanding is further compounded by how ReMoDetect constructs its training data. Unlike RLHF preference datasets that rely on pairwise human annotations, ReMoDetect labels LLM outputs as “preferred” and human-written texts as “dispreferred” without any human annotation. While useful for detection tasks, they differ substantially from the RLHF reward model’s training setup and may inadvertently blur the line between "human-preferred" and "human-written."
>
> We also appreciate the suggestion of comparing IRM score distributions across more fine-grained categories (e.g., preference vs. non-preference subsets for both LLM and human text). While insightful, such comparisons would require carefully constructed datasets containing both LLM-generated and human-written responses with explicit human preference annotations—which, to our knowledge, do not currently exist, as they would be prohibitively expensive and would undermine the scalability motivations behind RLHF.
>
> We hope this clarification helps reconcile the intuition and resolves the concern.
>
> > Weakness2: An additional benchmark dataset would help strengthen the paper, such as RAID. While DetectRL covers many scenarios, it would help support the papers claims if at least one other standard benchmark dataset suite is included.
>
> We sincerely appreciate the reviewer’s thoughtful suggestion to evaluate our method on an additional benchmark dataset such as RAID. We agree that demonstrating generalization across diverse benchmarks is important for validating the robustness of detection methods. In response, we have conducted comprehensive experiments on two benchmark in addition to DetectRL: RAID [1] and DivScore [2].
>
> We firstly evaluate our method on the RAID benchmark. Due to the large size of the dataset, we randomly sample 512 annotated pairs for each source model from the training split for evaluation. We adopt LLaMA-3.2-1B as the backbone detector, as it showed strong performance on DetectRL. We report AUROC as the main evaluation metric.
>
> | Method     | chatgpt | gpt4  | gpt3  | gpt2  | llama-chat | mistral | mistral-chat | mpt   | mpt-chat | cohere | cohere-chat |
> |-|-|-|-|-|-|-|-|-|-|-|-|
> | Log-Likelihood | 98.48   | 93.51 | 97.35 | 68.77 | 98.93      | 69.82   | 97.78        | 48.84 | 80.08    | 89.26  | 92.42       |
> | Log-Rank    | 98.71   | 92.95 | 96.94 | 71.13 | 99.12      | 71.34   | 98.03        | 51.83 | 81.27    | 87.43  | 91.89       |
> | Binoculars | 99.66   | 96.38 | 99.12 | 79.30 | 99.05      | 72.45   | 98.22        | 51.99 | 83.41    | 97.06  | 98.00       |
> | IRM        | 99.58   | 99.27 | 87.43 | 86.31 | 99.99      | 62.25   | 99.35        | 57.57 | 92.42    | 87.14  | 92.10       |
> | ReMoDetect | 99.69 | 97.82 | 94.57 | 57.36 | 99.33| 59.18 | 97.70| 43.06 | 89.63| 85.29  | 93.02  |
>
> As shown in the above Table, IRM achieves competitive performance across a wide range of models, demonstrating the generalizability of our approach beyond the original DetectRL setup.
>
> To further assess our method’s robustness in domain-specific and high-stakes settings, we additionally evaluate on DivScore [2], a recently released benchmark containing human preference data in legal and medical domains. It includes outputs from four reasoning-oriented LLMs: DeepSeek-R1, DeepSeek-V3, GPT-4o, and O3-mini. We use the official test split for evaluation and adopt LLaMA-3.2-1B as the backbone.
>
> | Method         | DeepSeek-R1 |       | DeepSeek-V3 |       | GPT-4o |       | O3-mini |       | Avg.  |       |
> |----------------|-------------|-------|-------------|-------|--------|-------|---------|-------|-------|-------|
> |                | AUROC       | F1    | AUROC       | F1    | AUROC  | F1    | AUROC   | F1    | AUROC | F1    |
> | Log-Likelihood | 88.64       | 86.11 | 98.87       | 97.68 | 85.96  | 81.50 | 90.37   | 87.47 | 90.96 | 88.19 |
> | Log-Rank       | 87.56       | 84.85 | 98.91       | 97.69 | 86.28  | 82.34 | 90.86   | 88.01 | 90.90 | 88.22 |
> | Binoculars     | 98.03       | 94.18 | 99.87       | 99.03 | 94.84  | 90.74 | 95.61   | 91.88 | 97.09 | 93.96 |
> | IRM            | 97.39       | 93.67 | 99.27       | 97.77 | 99.08  | 96.74 | 97.95   | 94.25 | 98.42 | 96.61 |
> | ReMoDtect      | 95.86       | 90.69 | 99.73       | 98.20 | 99.56  | 97.46 | 98.73   | 94.73 | 98.47 | 95.27 |
>
> | Method         | DeepSeek-R1 |       | DeepSeek-V3 |      | GPT-4o |       | O3-mini |      | Avg.  |      |
> |----------------|-------------|-------|-------------|------|--------|-------|---------|------|-------|------|
> |                | FPR         | FNR   | FPR         | FNR  | FPR    | FNR   | FPR     | FNR  | FPR   | FNR  |
> | Log-Likelihood | 21.44       | 9.19  | 3.23        | 1.50 | 15.00  | 19.02 | 17.48   | 9.11 | 14.29 | 9.71 |
> | Log-Rank       | 21.95       | 11.01 | 3.04        | 1.66 | 15.47  | 17.57 | 16.22   | 9.18 | 14.17 | 9.86 |
> | Binoculars     | 6.42        | 5.31  | 1.00        | 0.93 | 5.60   | 11.82 | 10.33   | 6.37 | 5.84  | 6.11 |
> | IRM            | 7.26        | 5.53  | 1.57        | 2.82 | 2.72   | 3.75  | 5.05    | 6.34 | 4.15  | 4.61 |
> | ReMoDtect      | 11.46       | 7.58  | 2.16        | 1.45 | 2.37   | 2.67  | 6.39    | 4.25 | 5.60  | 3.99 |
>
> Across both tables, IRM demonstrates consistently strong performance on recent reasoning models, achieving high AUROC and F1 while also maintaining low FPR and FNR. These results provide strong empirical evidence for the reliability and robustness of IRM when applied to real-world, safety-critical domains.
>
> We hope these additional results address the concern and further strengthen the empirical foundation of our work.
>
> > Weakness3: It would be helpful to understand the effect of LLM size on the detector performance, similar to [3] for DetectGPT. This would help users understand which instruction-tuned/base model policy pairs and their sizes are the most "universal" across LLMs and datasets.
>
> Thank you for raising this insightful point. We agree that understanding how the detector’s backbone size influences performance is important for identifying practical and robust detection strategies. To this end, we extend our analysis in Table 5 by evaluating IRM with larger (>7B) backbones. The table below reports the results on the DetectRL benchmark:
>
> | Backbone     | Avg.   |
> |--------------|-------|
> | Llama-3.2-1B | 91.77 |
> | Llama-3.2-3B | 91.08 |
> | Llama-3.1-8B | 82.31 |
> | Gemma-2-2B   | 77.05 |
> | Gemma-2-9B   | 64.34 |
>
> We observe a similar trend to that reported in [3]: smaller models within the same family often yield better detection performance. Notably, Llama-3.2-1B backbone achieves the best overall results, suggesting that smaller models may act as more universal detectors across various source models and datasets.
>
> > Weakness4: Is there a fully-supervised classifier baseline, such as RoBERTa? The fully supervised models seem trained for reward modeling, but I wonder how a baseline that directly tries to classify human/LLM text would compare.
>
> Thank you for raising this important point. DetectRL benchmark provides 4 supervised baselines including models such as RoBERTa and XLM-RoBERTa. We report the results of these supervised baselines and IRM:
>
> | Model              | Avg.  |
> |--------------------|-------|
> | RoBERTa-base       | 93.02 |
> | RoBERTa-Large      | 91.49 |
> | XLM-RoBERTa-base   | 91.71 |
> | XLM-RoBERTa-Large  | 90.59 |
> | IRM (Llama-3.2-1B) | 92.12 |
>
> We find that IRM outperforms 3 out of the 4 supervised baselines, despite requiring no task-specific training. This highlights the strength of our framework in a zero-shot setting.
>
> [1] RAID: A Shared Benchmark for Robust Evaluation of Machine-Generated Text Detectors
>
> [2] DivScore: Zero-Shot Detection of LLM-Generated Text in Specialized Domains
>
> [3] Smaller Language Models are Better Black-box Machine-Generated Text Detectors

---

> > ### Comment · Reviewer_xWbt · 2025-08-07
> >
> > Thank you for the thoughtful response. All my comments have been addressed, and I have raised my score.

---

### Decision · Program_Chairs · 2025-09-17

**Decision:**

Accept (poster)

**Comment:**

Summary of Claims and Findings:

This paper introduces the Implicit Reward Model (IRM), a novel, training-free detector for AI-generated text. It leverages the log-likelihood ratio between instruction-tuned and base models as a proxy for the reward signal, positing that AI text is over-optimized for this signal and is thus detectable. The method demonstrates state-of-the-art zero-shot performance and robustness across various models, domains, and adversarial attacks.

Strengths:

The paper's key strengths are its novelty and simplicity (the method is training-free), strong empirical performance that outperforms other zero-shot baselines, and insightful analysis of how factors like text length and detector size impact performance.

Weaknesses & Rebuttal Summary:

Initial weaknesses flagged by reviewers included a limited evaluation scope (a single benchmark, older models) and a lack of practical metrics like FPR/FNR. The authors' rebuttal was exemplary. They conducted extensive new experiments on the RAID and DivScore benchmarks, incorporated modern reasoning models (e.g., GPT-4o), and added the requested FPR/FNR analysis. This comprehensive response fully addressed all major reviewer concerns, leading to a consensus to accept.

Recommendation Justification:

The recommendation is based on the paper's novel and effective method, which was validated by an exceptionally thorough rebuttal. The new experiments significantly strengthened the paper's claims of robustness and generalizability. The work, in its revised form, is a strong and well-vetted contribution to a critical area of research.